# REPORT

# Nuclear poly-glutamine aggregates rupture the nuclear envelope and hinder its repair

Giel Korsten[1], Miriam Osinga[1], Robin A. Pelle[1], Albert K. Serweta[1], Baukje Hoogenberg[1], Harm H. Kampinga[2], and Lukas C. Kapitein[1,3]

**Huntington's disease (HD) is caused by a polyglutamine expansion of the huntingtin protein, resulting in the formation of polyglutamine aggregates. The mechanisms of toxicity that result in the complex HD pathology remain only partially understood. Here, we show that nuclear polyglutamine aggregates induce nuclear envelope (NE) blebbing and ruptures that are often repaired incompletely. These ruptures coincide with disruptions of the nuclear lamina and lead to lamina scar formation. Expansion microscopy enabled resolving the ultrastructure of nuclear aggregates and revealed polyglutamine fibrils sticking into the cytosol at rupture sites, suggesting a mechanism for incomplete repair. Furthermore, we found that NE repair factors often accumulated near nuclear aggregates, consistent with stalled repair. These findings implicate nuclear polyQ aggregate-induced loss of NE integrity as a potential contributing factor to Huntington's disease and other polyglutamine diseases.**

## Introduction

Huntington's disease (HD) is a debilitating neurodegenerative disease caused by a CAG repeat expansion in exon-1 of the huntingtin gene (Bates et al., 2015), resulting in the expression of polyglutamine (polyQ) proteins that are prone to aggregate. A unified understanding of HD pathology is lacking, but polyQ aggregation has been associated with a wide range of cellular defects, including reduced protein quality control and transcriptional deregulation (Cortes and La Spada 2014; Hay et al., 2004; Hipp et al., 2012; Park et al., 2013; Sugars and Rubinsztein 2003), as well as a compromised barrier function of the nuclear envelope (NE) (Gasset-Rosa et al., 2017; Grima et al., 2017). While the latter has primarily been attributed to impaired nucleocytoplasmic shuttling due to the sequestration of shuttling factors (Bitetto and Di Fonzo 2020; Gasset-Rosa et al., 2017; Grima et al., 2017), cells with nuclear polyQ aggregates also display abnormalities in their nuclear lamina (Chapple et al., 2008; Gasset-Rosa et al., 2017; Grima et al., 2017), an intranuclear intermediate filament scaffold that serves to protect the NE from rupturing (Gruenbaum and Foisner 2015). This suggests that ruptures of the NE might contribute to impaired nuclear barrier function in HD. Consistently, recent work has revealed that polyQ aggregates can directly disrupt organelle membranes (Bäuerlein et al., 2017; Riguet et al., 2021) and interact with the NE (Chapple et al., 2008; Liu et al., 2015; Lu et al., 2015; Waelter et al., 2001). However, whether such interactions result in polyQ

aggregate-induced ruptures that compromise the barrier function of the NE has remained unexplored.

To study the effect of polyQ aggregates on NE integrity, we expressed various forms of huntingtin exon1 (Hageman et al., 2010) in cells stably expressing RFP with a nuclear localization signal (U2OS-RFP-NLS; Fig. 1 A) (Vargas et al., 2012). As expected, an expanded form of huntingtin targeted to the nucleus (polyQ74-NLS) exclusively formed nuclear aggregates, while non-expanded huntingtin (polyQ23-NLS) formed none. Expression of expanded, non-targeted huntingtin (polyQ74) only resulted in the formation of cytoplasmic inclusions in this cell line (Fig. 1 B). We then performed long-term (8 h) live-cell imaging of these cells and found that cells with nuclear aggregates frequently showed loss of NE integrity (32.9 ± 8.6%, n = 325 cells; Fig. 1, C–E and Video 1), demonstrated by a rapid loss of RFP from the nucleus. Single cells often showed multiple rounds of rupture (1.8 ± 1.3 ruptures per cell, n = 108 cells) and repair, indicated by the reaccumulation of RFP in the nucleus (Fig. S1, A–D). In contrast, expression of either cytosolic or non-expanded polyQ protein only resulted in a minor increase in NE ruptures (7.7 ± 8.3% and 7.2 ± 1.9%, n = 165 and 506 cells) compared to control (2.0 ± 0.8%, n = 1,069 cells; Fig. 1 E). Ruptures in polyQ74-NLS–expressing cells were often preceded by NE blebbing events (41.9 ± 4.1%, n = 107 ruptures; Fig. 1, C, F and G; and Fig. S1, B and E–G), similar to the NE herniations shown to

[1]Cell Biology, Neurobiology and Biophysics, Department of Biology, Faculty of Science, Utrecht University, Utrecht, Netherlands;   [2]Department of Cell Biology, University Medical Center Groningen, University of Groningen, Groningen, Netherlands;   [3]Centre for Living Technologies, Alliance TU/e, WUR, UU, UMC Utrecht University, Utrecht, Netherlands.

Correspondence to Lukas C. Kapitein: l.kapitein@uu.nl.



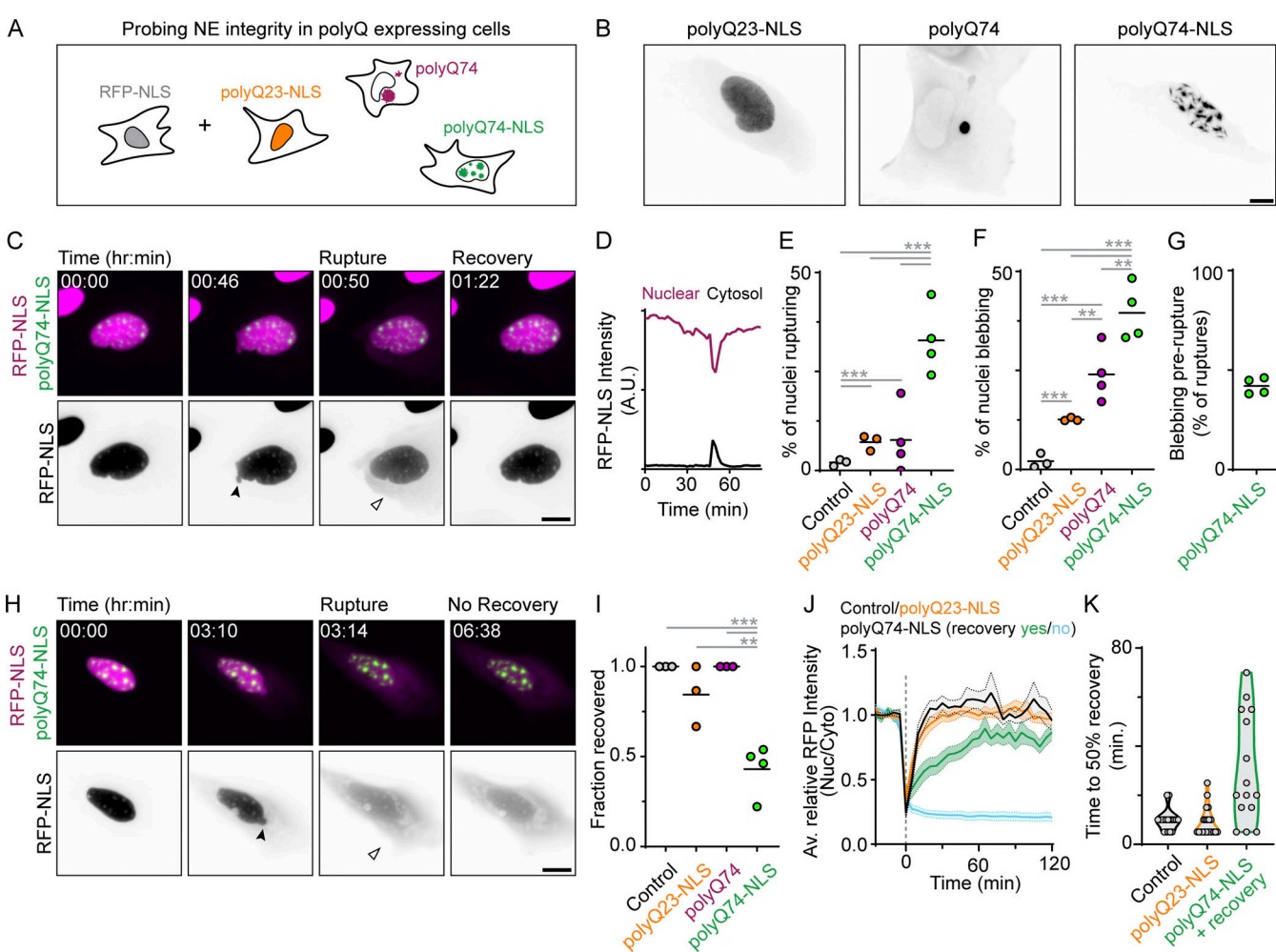

Figure 1. **Nuclear aggregates induce NE rupture and impair recovery. (A)** Schematic representation of experimental design probing NE integrity in U2OS-RFP-NLS (gray) cells expressing nuclear (polyQ74-NLS), non-expanded (polyQ23-NLS), or non-targeted (polyQ74) polyQ protein. **(B)** Representative images of localization and aggregate formation in cells expressing various polyQ constructs. **(C)** Time-lapse images of NE rupture in representative U2OS-RFP-NLS (magenta) cell expressing polyQ74-NLS (green), resulting in transient loss of nuclear RFP-NLS enrichment. Solid arrowhead marks nuclear bleb, open arrowhead marks cytosolic RFP-NLS. **(D)** Nuclear (magenta curve) and cytosolic (black curve) RFP-NLS intensity of cell in C. **(E and F)** Percentage of cells showing NE rupture (E) and NE blebbing (F) in U2OS-RFP-NLS control cells and cells expressing polyQ23-NLS, polyQ74 and polyQ74-NLS ($n_{cells}$ = 1,069, 506, 163, 325; $N$ = 3, 3, 4, 4). Dots represent independent replicates. **(G)** Percentage of ruptures in polyQ74-NLS expressing cells that were preceded by blebbing events ($n$ = 107 ruptures; $N$ = 4). **(H)** Time-lapse images of NE rupture in representative U2OS-RFP-NLS cells expressing polyQ74-NLS, showing permanent loss of RFP-NLS enrichment after NE rupture. **(I)** Fraction of U2OS-RFP-NLS cells that recovered nuclear enrichment of RFP-NLS signal after NE rupture ($n$ = 21, 37, 13, 103; $N$ = 3, 3, 4, 4). **(J)** Recovery of normalized nuclear RFP-NLS enrichment after rupture ($t$ = 0) in control cells (black curve) and cells expressing polyQ23-NLS (orange curve) and polyQ74-NLS cells with (green curve) or without recovery (cyan curve; $n$ = 18, 21, 17, 31; $N$ = 3, 3, 4, 4). **(K)** Graph of time until 50% recovery after NE rupture of individual control cells or cells expressing polyQ23-NLS or polyQ74-NLS shown in J. Horizontal bars represent mean ± SEM. Scale bars are 10 μm (B) or 15 μm (C and G). *P < 0.05, **P ≤ 0.01, ***P ≤ 0.001, assessed by Fischer's exact test.

arise during constricted migration or after lamin depletion (Denais et al., 2016; Hatch and Hetzer 2016; Raab et al., 2016). While NE blebbing also occurred at higher frequency in cells expressing non-expanded polyQ23-NLS or cytosolic aggregates (12.6 ± 0.4% and 24.0 ± 6.8%), they were most frequent in cells with nuclear aggregates (39.6 ± 7.0%; Fig. 1 F). Interestingly, cytosolic aggregates close to the NE frequently deformed the nucleus, but without causing ruptures (Fig. S1, H–J). These findings demonstrate that nuclear polyQ aggregates induce NE blebbing and rupture, reflecting a compromised barrier function of the NE.

While NE ruptures are typically quickly repaired, as revealed by the reaccumulation of RFP-NLS (Denais et al., 2016; Hatch and Hetzer 2016; Vargas et al., 2012; Xia et al., 2018), we noticed multiple instances of permanent loss of NE integrity after the rupture in cells with nuclear aggregates (Fig. 1 H; Video 2; and Fig. S1, A–C). To determine whether this impaired recovery was specific for cells with nuclear aggregates, we scored the fraction of cells that recovered after NE rupture and determined their recovery dynamics. Indeed, ruptures in cells with nuclear aggregates recovered less often (43 ± 14% recovery; Fig. 1 I) and recovered slower than in control cells or cells expressing non-

expanded polyQ protein (Fig. 1, J–K). These results suggest that nuclear aggregates interfere with NE resealing.

To test whether the prolonged loss of NE integrity in the presence of aggregates was caused indirectly through initiation of cell death, we imaged U2OS-RFP-NLS cells expressing polyQ74-NLS aggregates and used Mitoview as a live-cell marker for cell viability. Mitoview fluorescence is dependent on mitochondrial membrane potential ($\Delta\Psi m$), which is lost upon initiation of cell death (Ricci et al. 2003). We found that the majority of cells that displayed unhealed ruptures did not lose $\Delta\Psi m$ at any time during imaging (71.4 ± 13.1%; Fig. S1, K and M). Only a small fraction of unhealed ruptures was preceded by $\Delta\Psi m$-loss (13.5 ± 5.8%), whereas some cells showed loss of $\Delta\Psi m$ during or after NE rupture (5.6 ± 4.8% and 9.5 ± 9.6%, Fig. S1, L and M). These findings show that cell viability is not acutely lost in cells with unhealed NE ruptures and imply that the contribution of cell death pathways to our observations is likely limited.

Since both NE ruptures and impaired recovery were specifically induced by nuclear aggregates, we hypothesized that these aggregates would locally deform and disrupt the nuclear lamina and NE. First, we determined whether ruptures indeed occurred close to nuclear aggregates. To this end, we co-expressed polyQ74-NLS with mCherry-tagged guanosine 3′,5′-monophosphate–adenosine3′,5′-monophosphate (cyclic GMP-AMP) synthase (cGAS), a DNA binding protein rapidly recruited to sites of NE rupture (Civril et al., 2013; Xia et al., 2018) (Fig. 2 A). Indeed, ruptures occurred specifically near nuclear aggregates, as evidenced by the accumulation of endogenous (Fig. S2 A) and mCherry-cGAS around aggregates (Fig. 2, B and C; Video 3; and Fig. S2, B and C). In some cells, mCherry-cGAS accumulated around multiple distinct aggregates at different times during imaging, suggesting that the ruptures previously observed in U2OS-RFP-NLS cells could have occurred at different sites (Fig. S2, B and C; and Fig. S1, A–D). We then used mCherry-cGAS as a marker to validate the occurrence of aggregate-induced ruptures in non-transformed RPE-1 cells, and also frequently found ruptures at nuclear aggregates in this cell-type (Fig. S2 D). Furthermore, by using far-red fluorescent emiRFP-cGAS in aggregate-expressing U2OS-RFP-NLS cells, we could directly demonstrate that the NE blebs emerging in cells with nuclear aggregates were also the sites of rupture (Fig. 2, D and E).

Next, we tested whether these sites of aggregate-induced rupture also display lamina deformations. We expressed nuclear aggregates and HaloTag-lamin A in our reporter cells and found that lamin A intensity was often reduced near nuclear aggregates, but rapidly accumulated at aggregates upon NE rupture (Fig. 2, F and G; Video 4; and Fig. S2, E and F). This accumulation of lamin A partly resembles the "scar" formation following rupture that was previously reported and hypothesized to be locally protective (Denais et al., 2016; Isermann and Lammerding 2017). Following aggregate-induced rupture, however, lamin A accumulation sometimes seemed incomplete and disrupted by the presence of the aggregate (Fig. 2 G). We also observed occasional accumulation of HaloTag-lamin A at nuclear aggregates prior to NE rupture, possibly reflecting the reinforcement of lamina sites that were destabilized by aggregates (Fig. S2, G and H). To validate and quantify the impact

of aggregates on the nuclear lamina, we used immunostaining of endogenous lamin B1 and analyzed nuclear deformations in different conditions (Fig. 2 H). We focused on endogenous lamin B1 because it does not display scar formation (Denais et al., 2016; Kono et al., 2022) and therefore enables us to isolate destabilization from repair and reinforcement. While cytosolic aggregates often induced a single, large nuclear deformation (27.6 ± 4.9%, n = 253 cells; Fig. 2, H and I), nuclear aggregates induced dissociation of parts of the lamin B1 meshwork, resulting in areas devoid of lamin B1 (35.0 ± 2.4%, n = 289 cells; Fig. 2, H and J) that are likely prone to NE rupture (Denais et al., 2016). Interestingly, these disruptions were only rarely present in cells with cytosolic aggregates (12.8 ± 2.3%, n = 253 cells; Fig. 2, H and J). Together, these findings indicate that nuclear aggregates locally disrupt the nuclear lamina, which subsequently leads to frequent NE blebbing. Alternatively, the presence of nuclear aggregates might have other, more indirect effects on NE stability, for example via systemic disruption of chromatin or altered nucleocytoplasmic shuttling. Nonetheless, the defects that we observed are all important indicators of NE weakening and rupture propensity (Denais et al., 2016; Earle et al., 2020; Raab et al., 2016; Xia et al., 2018) and likely underly the rupture induction at nuclear aggregates.

While the propensity to NE rupture arises from defects in the nuclear lamina (Denais et al., 2016; Earle et al., 2020; Raab et al., 2016; Xia et al., 2018), previous work has emphasized the contribution of contractile forces acting upon the nucleus to NE rupture induction (Hatch and Hetzer 2016). Nuclei are exposed to cytoskeletal forces that are transduced from the actin cytoskeleton through the Linker of Nucleus and Cytosol (LINC) complex (Fig. 3 A) (Crisp et al., 2006; Hatch and Hetzer 2016; Padmakumar et al., 2005). We hypothesized that these forces acting upon the nucleus are also contributing to rupture induction at aggregate-induced lamina weak spots. Therefore, we treated U2OSWT cells expressing polyQ74-NLS aggregates and rupture marker mCherry-cGAS with blebbistatin, a myosin II inhibitor (Fig. 3 A), to disrupt actin contractility and found a ~43.0% reduction in the amount of cells with NE ruptures compared with control (33.6 ± 4.6% versus 19.2 ± 2.1%, n = 1,042 and 978; Fig. 3, B and C). These results support a model for polyQ aggregate-induced ruptures in which lamina and NE destabilization increase rupture propensity, while pressure applied to the nucleus contributes to the ultimate induction of ruptures.

Because NE ruptures appear to occur in close proximity to nuclear aggregates, we set out to perform a nanoscale analysis of local deformations and disruptions near nuclear aggregates. For this, we turned to 10-fold robust expansion microscopy (TREx), which enables specific labeling of aggregates and lamina in combination with visualization of the NE membranes (Damstra et al., 2022). Expanded cells displaying different degrees of aggregation revealed a striking improvement in resolution compared with confocal microscopy (Fig. 4 A and Video 5), allowing three-dimensional visualization and segmentation of individual nuclear aggregates. These aggregates had dense cores with individual fibrils protruding outward (Fig. 4 B). Cytosolic polyQ74 aggregates showed a similar architecture, especially for smaller

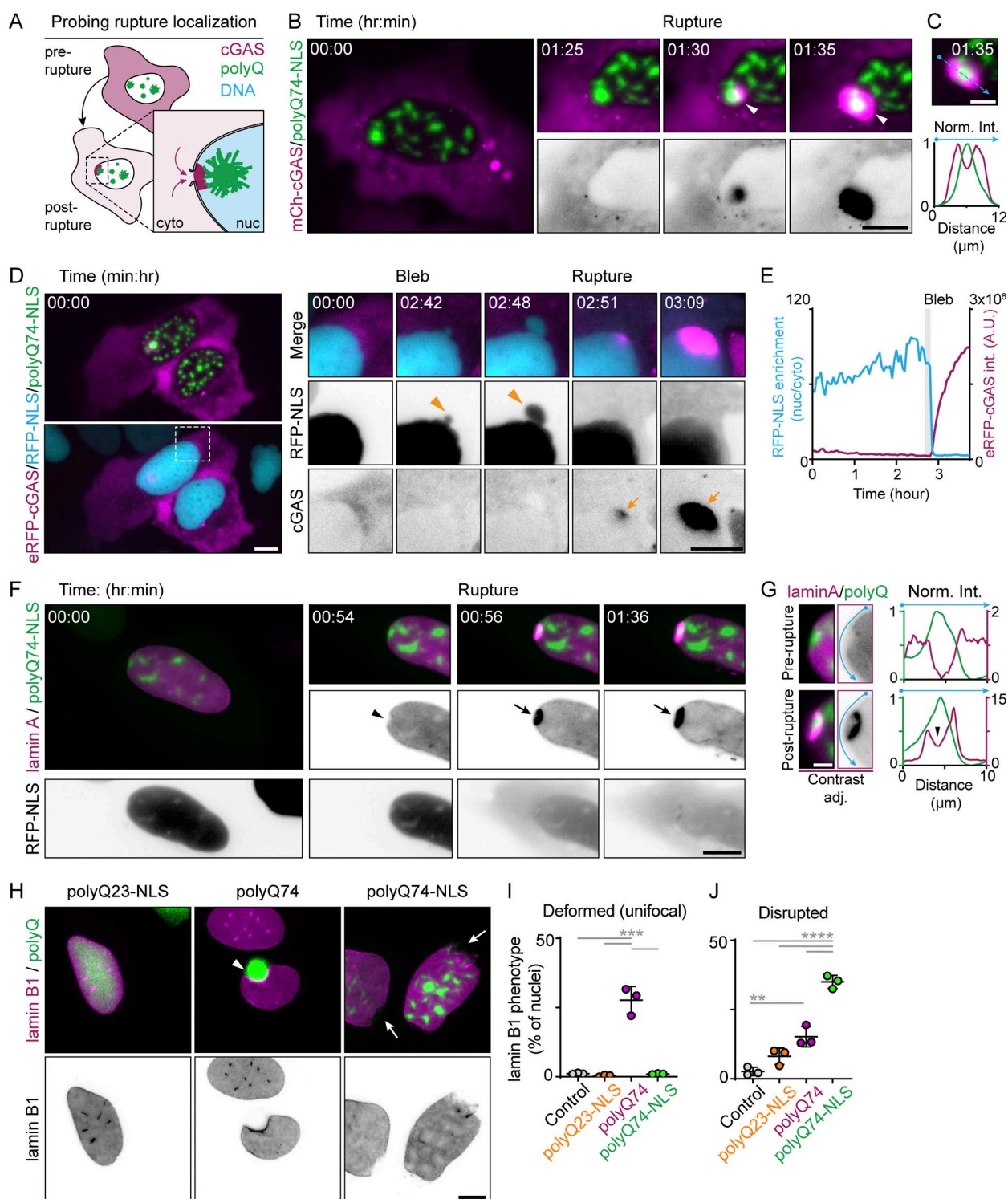

Figure 2. **Nuclear aggregates induce local disruptions of the nuclear envelope and nuclear lamina. (A)** Schematic representation of the use of DNA sensing cGAS expression as a tool for detecting NE rupture localization. **(B)** Time-lapse images of representative U2OS cell expressing cGAS (magenta) and polyQ74-NLS (green), showing nuclear cGAS entry (white arrowhead) around a nuclear aggregate. **(C)** Intensity profile of cGAS accumulation around aggregate shown in B. **(D)** Time-lapse images of representative U2OS-RFP-NLS cell (cyan) expressing polyQ74-NLS (green) and eRFP-cGAS (emiRFP670; magenta). Zooms show blebbing (orange arrowhead) and subsequent eRFP-cGAS accumulation (orange arrows). **(E)** Graph depicting increase in eRFP-cGAS signal (magenta curve) after rupture indicated by loss of RFP-NLS enrichment (cyan curve). Duration of the blebbing event is shown in gray. **(F)** Live-cell imaging of representative U2OS-RFP-NLS (gray) cells expressing polyQ74-NLS (green) and HaloTag-lamin A (magenta) showing local lamin A depletion (arrowhead) and scar formation post-rupture (black arrowhead). **(G)** Zooms and intensity profiles along the NE of rupture site shown in F pre- and post-rupture. **(H)** Representative images of U2OS cells expressing polyQ23-NLS, polyQ74-NLS, or polyQ74 (green) immunostained for lamin B1 (magenta), showing lamin B1 disruption (white arrows) or unifocal (white arrowheads) lamin B1 deformation. **(I and J)** Percentage of U2OS control cells, or cells expressing polyQ23-NLS,

polyQ74, or polyQ74-NLS that show unifocal deformation (I) or disruption (J) of lamin B1. Horizontal bars represent mean ± SD and dots represent independent replicates ($n_{cells}$ = 589, 480, 253, 289; $N$ = 3). PolyQ (B, C, and F) and cGAS (B and C) signals were gamma-adjusted (γ = 0.75). Scale bars represent 10 µm (B, D, F, and H) and 5 µm ($B_{zoom}$).**P ≤ 0.01, ***P ≤ 0.001, ****P ≤ 0.0001, assessed by Fischer's exact test (I) or one-way ANOVA with Tukey's multiple comparisons test (J).

aggregates (Fig. S3, A and B). And while large aggregates had substantially larger cores, polyQ fibrils could still be found protruding from these inclusions (Fig. S3 B). Together, these observations resemble earlier observations of polyQ aggregates using electron microscopy (Bäuerlein et al., 2017; Riguet et al., 2021).

Next, labeling of nuclear aggregates and endogenous lamin B1 allowed us to observe the structure of the lamin B1 meshwork using TREx (Fig. S3, C–E). In control cells, lamin B1 was present as a continuous meshwork lining the nuclear membrane, resembling earlier results obtained using other super-resolution techniques (Nmezi et al., 2019; Shimia et al., 2015; Stiekema et al., 2021) (Fig. S3, C–E). Consistent with our earlier data (Fig. 2, H–J), cells with nuclear aggregates displayed strong lamin B1 abnormalities. Nuclear aggregates colocalized with large disruptions (Fig. 4, C and D), as well as smaller disruptions where polyQ fibrils appeared to protrude through holes in the lamina meshwork (Fig. 4, C and E; and Fig. S3 F). In contrast, the lamin B1 meshwork appeared intact near cytosolic aggregates, consistent with earlier reports (Bäuerlein et al., 2017; Riguet et al., 2021) (Fig. S3 G). Using a total membrane stain (mCLING), we then visualized the NE in cells with or without nuclear aggregates. In control cells, the NE appeared as a continuous structure clearly distinct from other cellular membranes (Fig. S3 H). However, in cells expressing polyQ74-NLS, we found multiple instances of rupture sites (identified by intranuclear mCherry-cGAS accumulation), where membrane deformations and disruptions were apparent around nuclear aggregates (Fig. 4 F; and Fig. S3, I and J). Such deformations near NE rupture sites are reminiscent of the accumulation of membrane found as a result of prolonged NE remodeling by the endosomal sorting complex-III (ESCRT-III), which also induces long-term loss of NE integrity (Vietri et al., 2020).

Normally, efficient NE repair is regulated by the transient recruitment of various proteins including ESCRT-III, BAF (barrier-to-autointegration factor), and LEM-domain (Lap2, emerin,

**Figure 3. Inhibition of actin contractility reduces polyQ aggregate-induced rupture frequency. (A)** Schematic representation of blebbistatin inhibition of myosin II resulting in reduced force transduction onto the nucleus. **(B)** Representative images of U2OS cells expressing polyQ74-NLS (green) and mCherry-cGAS (magenta) and treated with DMSO or blebbistatin. Orange arrowheads indicate nuclear mCherry-cGAS accumulation. **(C)** Quantification of the percentage of cells in B showing nuclear mCherry-cGAS accumulation after treatment with DMSO or blebbistatin ($n_{cells}$ = 1,042, 978; $N$ = 3). Dots represent averages of independent replicates grouped by shades of gray. Horizontal bars represent the mean. Scale bar represents 20 µm. ****P ≤ 0.0001, assessed by Fischer's exact test.

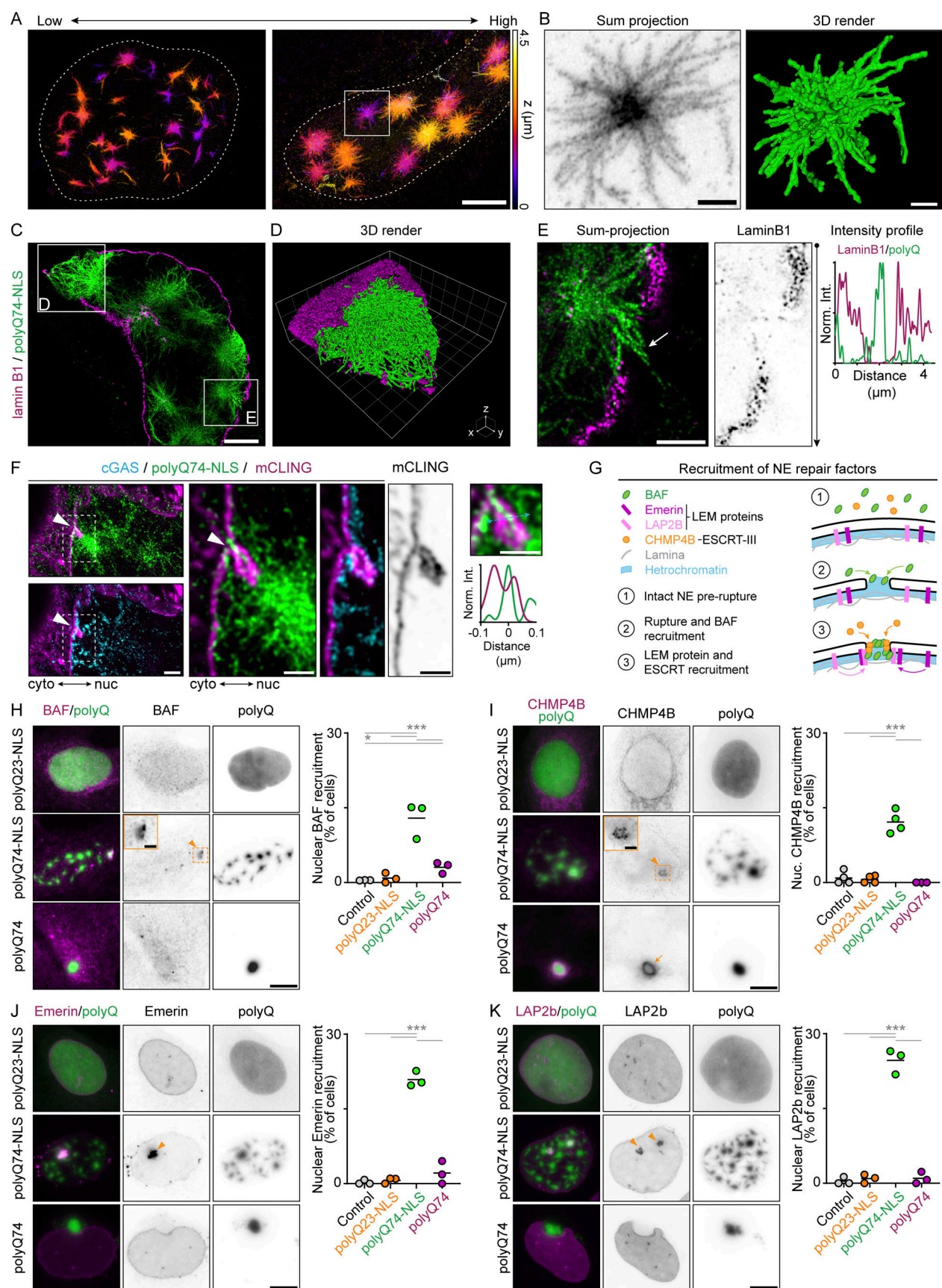

Figure 4. **LEM domain proteins and NE repair factors accumulate near nuclear polyQ aggregates. (A)** Depth-encoded color projection of representative expanded U2OSWT cells expressing polyQ74-NLS, showing low and high stages of aggregation progression. See also Video 5. **(B)** Sum projection (inverted

grayscale) and 3D volumetric render (green) of single aggregates shown in A. **(C)** Representative image of expanded U2OSWT cell expressing polyQ74-NLS (green), stained for lamin B1 (magenta). **(D)** 3D volumetric render of lamin B1 disruption shown in C. **(E)** Sum projection and intensity profile of cell in C, showing polyQ74-NLS fibrils protruding through a lamin B1 disruption (white arrow). **(F)** Representative images of expanded U2OSWT cell expressing polyQ74-NLS (green) and cGAS (cyan), stained with total membrane stain (mCling; magenta). White arrowheads indicate local nuclear membrane deformation at a nuclear aggregate-induced rupture site (marked by cGAS accumulation). **(G)** Schematic representation of pre-rupture localization (1) and recruitment of repair factors after NE rupture. Upon NE rupture BAF (2), and LEM domain proteins and CHMP4B (3) are recruited to the gap in the NE. **(H–K)** Representative images of U2OSWT cells expressing polyQ23-NLS, polyQ74-NLS or polyQ74 (green) immunolabeled with various antibodies (magenta). Graphs show quantification of the percentage of control cells, or cells expressing polyQ23-NLS, polyQ74-NLS or polyQ74 that show nuclear accumulation of BAF (H; $n_{cells}$ = 908, 722, 823, 384, respectively; $N$ = 3), CHMP4B (I; $n_{cells}$ = 313, 306, 269, 121; $N$ = 4,4,4,3), Emerin (J; $n_{cells}$ = 384, 307, 444, 306; $N$ = 3) or LAP2B (K; $n_{cells}$ = 461, 332, 437, 352; $N$ = 3). Dots represent independent replicates. Orange arrowheads (H–K) and zooms (H and I) indicate nuclear accumulation of immunolabeled protein at nuclear aggregates. Orange arrows indicate co-aggregation of CHMP4B around cytosolic aggregate (I). Contrast of GFP signal was adjusted for (H–K) to aid visibility. Scale bars represent 2.5 μm (A), 0.5 μm (B, C, and F), 0.25 μm ($F_{zoom}$), 1 μm (E), and 10 μm (H–K). Voxel sides are 600 nm (grid of D). Horizontal bars represent mean. *P < 0.05, ***P ≤ 0.001, assessed by Fischer's exact test.

man1) proteins (Fig. 4 G) (Denais et al., 2016; Halfmann et al., 2019; Lusk and Ader 2020; Raab et al., 2016; Young et al. 2020). Membrane resealing is thought to occur through the assembly and disassembly of ESCRT-III complex subunits such as CHMP4B (Denais et al., 2016; Lusk and Ader 2020; Raab et al., 2016), but is likely preceded by the recruitment of cytoplasmic BAF to exposed chromatin, especially when ruptures are larger (Lusk and Ader 2020). BAF in turn facilitates the accumulation of LEM-domain proteins, like emerin and LAP2b, that are located in the INM (Halfmann et al., 2019). Together, these proteins link the NE to lamins and chromatin and form a template for NE reformation (Fig. 4 G).

We wondered whether the presence of nuclear polyQ aggregates would disrupt the localization of NE-repair factors. We therefore probed the endogenous localization of BAF, CHMP4B, emerin, and LAP2B in control cells and in cells expressing the various forms of polyQ used previously. In control cells and cells expressing polyQ23-NLS or polyQ74, BAF and CHMP4B were localized primarily to the cytoplasm (Fig. 4, H and I), while emerin and LAP2b showed clear NE localization (Fig. 4, J and K). However, in cells with nuclear aggregates, we frequently observed nuclear accumulation of BAF around nuclear aggregates (13.1 ± 3.6%). Such nuclear foci were only rarely present in control cells (0.5 ± 0.1%) and cells expressing non-aggregated polyQ23-NLS or polyQ74 cytosolic aggregates (0.9 ± 1.0% and 3.1 ± 1.2%, respectively; Fig. 4 H). Interestingly, ESCRT-III member CHMP4B also frequently formed foci that colocalized with nuclear polyQ aggregates (12.1 ± 2.2%). Control cells and polyQ23-NLS expressing cells only showed such foci infrequently (0.9 + 1.2% and 0.6 ± 0.7%, respectively; Fig. 4 I). Importantly, while cytosolic aggregates did colocalize with cytosolic CHMP4B, possibly due to local disruption of endomembranes or co-aggregation (Bąk and Milewski 2010; Bäuerlein et al., 2017), cells with cytosolic aggregates did not show nuclear CHMP4B foci (Fig. 4 I). Notably, these results not only suggest higher rupture frequencies induced by nuclear aggregates, consistent with our live-cell observations of ruptures in U2OS-RFP-NLS cells, but might also reflect ESCRT-III stalling at rupture sites. Such prolonged presence of ESCRT-III could explain the membrane deformations found at rupture sites (Fig. 4 F) (Vietri et al., 2020).

Next, endogenous labeling of emerin and LAP2b revealed that the presence of nuclear aggregates frequently induced the formation of foci resembling those found for BAF and CHMP4B

(emerin: 20.9 ± 1.5%; LAP2b: 24.7 ± 2.5%; Fig. 4, J and K). This was in contrast to findings in control cells and cells expressing polyQ23-NLS or cytosolic aggregates since they did not show increased amounts of nuclear emerin (0.3 ± 0.4%, 0.7 ± 0.6% and 2.1 ± 2.3%, respectively) or LAP2b foci (0.4 ± 0.7%, 0.9 ± 0.9% and 1.0 ± 1.2%, respectively). These accumulations of LEM-domain proteins could indicate ongoing repair (Young et al. 2020), but could also be a consequence of the local hindering of proper NE (re)assembly by the presence of polyQ fibrils (Fig. 4, F and I). Overall, the reduced frequency and efficiency of NE repair, in combination with nuclear envelope deformations and the accumulation of repair factors at rupture sites suggests that nuclear polyQ aggregates could indeed interfere with proper membrane resealing after NE rupture.

Huntington's disease primarily affects neurons in various brain regions (DiFiglia et al., 1997; Gutekunst et al., 1999; Seidel et al., 2016). A decline in NE barrier function is especially relevant in neuronal cells since postmitotic neurons must ensure the maintenance of a distinct and long-lived NE (Alcalá-Vida et al., 2021; D'Angelo et al., 2009) and rupture induction might have accumulating effects in these cells. Therefore, we expressed expanded or non-expanded polyQ protein in primary rat hippocampal neurons. Coexpression of mCherry-cGAS allowed for visualization of nuclear rupture sites in these cells. As expected, expression of non-expanded, polyQ23-NLS protein in neurons did not result in aggregate formation (Fig. 5 A), and only a small portion of polyQ23-NLS expressing cells showed cGAS accumulation in the nucleus (5.1 ± 2.0%, $n$ = 519 cells; Fig. 5 B). Similar to expression in U2OS cells, neurons expressing polyQ74-NLS exclusively formed intranuclear aggregates (Fig. 5 A). Interestingly, in contrast to the exclusively cytoplasmic localization in U2OS cells, expression of non-targeted polyQ74 in neurons resulted in the formation of both cytosolic and nuclear aggregates in the majority of cells (58.3 ± 3.8%, $n$ = 264 cells; Fig. 5 A), resembling the localization of native non-targeted polyQ aggregates in HD (Davies et al., 1997; DiFiglia et al., 1997; Gutekunst et al., 1999). These differences in the localization of polyQ74 might be caused by early mitotic activity of U2OS cells, since mitosis might remove aggregate species from the nucleus, but could also be the result of functional differences in nuclear protein quality control systems between U2OS and neuronal cells. While high-resolution imaging of nuclear polyQ fibrils in neurons proved challenging due to lower throughput and labeling density, we were able to visualize fibrillar cytosolic

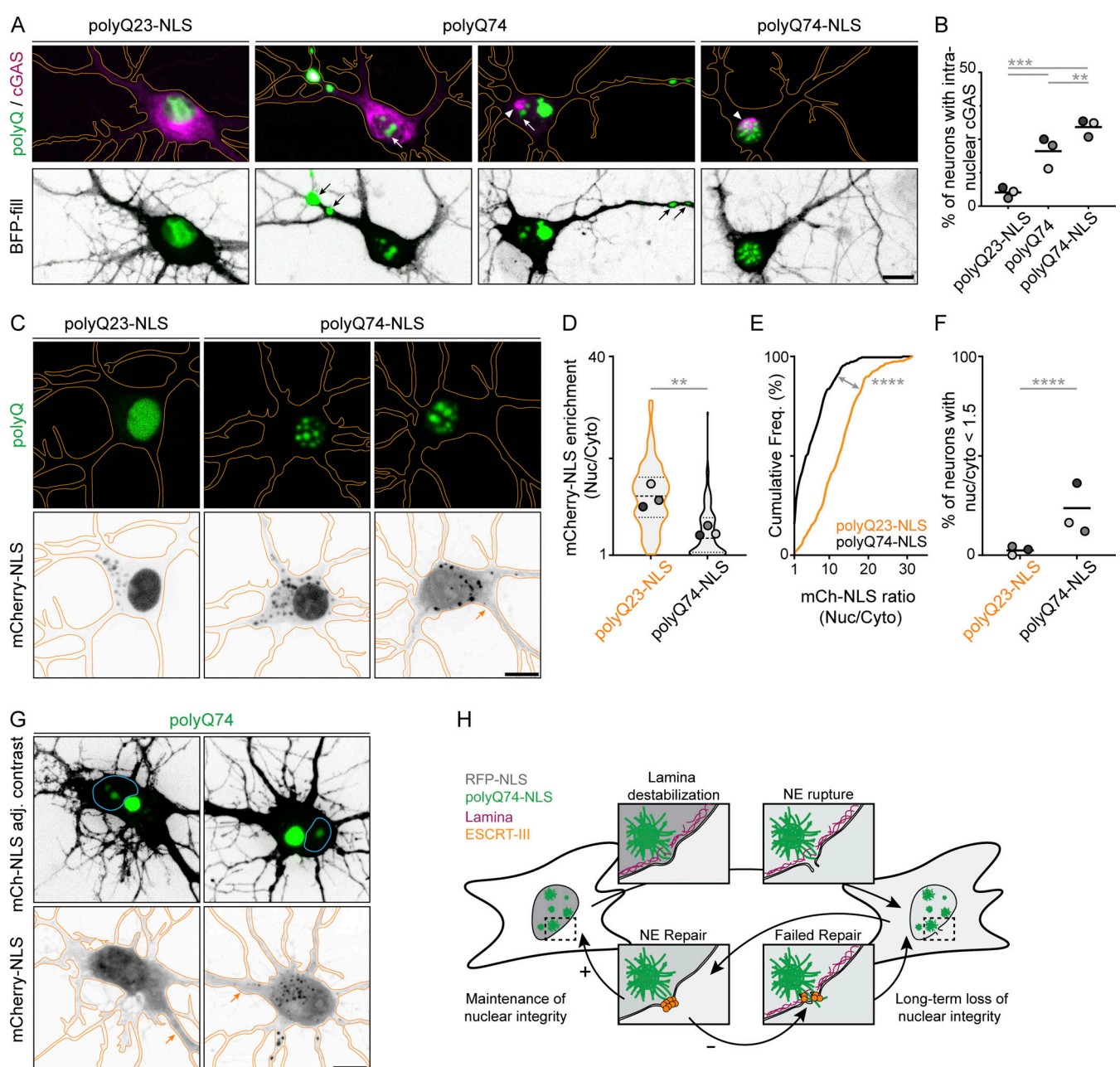

Figure 5. **Nuclear polyQ aggregates induce sustained NE ruptures in primary rat hippocampal neurons. (A)** Representative images of primary hippocampal neuron expressing BFP-fill (inverted grayscale), mCherry-cGAS (magenta), and polyQ23-NLS, polyQ74, or polyQ74-NLS (green). Neurons expressing polyQ74 show aggregation in soma, dendrites (black arrows), and nucleus (white arrows). Redistribution of mCherry-cGAS to the nucleus indicates NE rupture (white arrowhead). **(B)** Quantification of the percentage of neurons shown in A showing intranuclear cGAS accumulation (polyQ23-NLS, polyQ74, polyQ74-NLS; $n$ = 519, 264, 388; $N$ = 3). **(C)** Representative images of primary hippocampal neurons expressing polyQ23-NLS or polyQ74-NLS (green) and mCherry-NLS (gray) showing loss of nuclear mCherry-NLS enrichment (orange arrow). **(D–F)** Graphs depict nuclear enrichment of mCherry-NLS signal (D), cumulative frequency distribution of mCherry-NLS ratios (nuclear/cytoplasm; E), and the percentage of neurons with low mCherry-NLS ratio (F; $n_{polyQ23-NLS}$ = 316 and $n_{polyQ74-NLS}$ = 259 cells, $N$ = 3). **(G)** Representative images of hippocampal neurons expressing mCherry-NLS (inverted grayscale and contrast adjusted) and polyQ74 (green) that show aggregation in the cytosol and nucleus (outlined in cyan). Orange arrow indicates loss of nuclear mCherry-NLS enrichment. **(H)** Proposed model for lamin disruption and sustained NE rupture induced by nuclear aggregates. Failure to restore nuclear integrity could lead to sustained loss of nuclear integrity and prolonged accumulation of NE repair factors at rupture sites. Neurons are outlined in orange. PolyQ74-NLS and polyQ74 signal was gamma adjusted ($\gamma$ = 0.75; A and G). All scale bars indicate 10 μm. Dots represent averages of independent replicates grouped by shades of gray. *P <0.05, **P ≤ 0.01, ***P ≤ 0.001, ****P ≤ 0.0001, assessed by Fischer's exact test (B and F), unpaired Student's $t$ test (D), or Mann–Whitney test (E).

polyQ74 aggregates in neurons using expansion microscopy (Fig. S3 K). We found a strong increase in the amount of rupture sites in neurons with polyQ74 (~4.0-fold, 20.6 ± 5.8%, n = 264 cells) and polyQ74-NLS (~5.8-fold, 29.6 ± 3.2%, n = 388 cells) aggregates compared with polyQ23-NLS controls (Fig. 5 B).

We then wondered whether ruptures can also lead to long-term loss of NE integrity in neurons. However, due to photo-toxicity induced by long-term imaging and low transfection efficiencies, robust quantification of ruptures using live neurons proved challenging. We, therefore, opted to transfect neurons with polyQ23-NLS or polyQ74-NLS and mCherry-NLS as a nuclear rupture marker and imaged these cells at a single time point ~48 h after transfection (Fig. 5 C). Interestingly, neurons with nuclear polyQ74-NLS aggregates showed significantly lower ratios of nuclear/cytoplasmic mCherry-NLS intensity compared with polyQ23-NLS expressing neurons (~2.2-fold reduction, n = 259 and 316 cells, respectively; Fig. 5, C–E). This reduced enrichment found in polyQ74-NLS expressing neurons could be explained not only by ruptures that are still being repaired but could also be a result of polyQ-mediated interference with nucleocytoplasmic shuttling (Alcalá-Vida et al., 2021; Gasset-Rosa et al., 2017). More importantly, however, we found a substantial fraction of neurons with nuclear aggregates that showed no clear nuclear enrichment at all (23.7 ± 11.0%), while this phenotype was largely absent in neurons expressing polyQ23-NLS (2.4 ± 2.2%; Fig. 5, C and F). Several cells expressing non-targeted polyQ74 that displayed nuclear aggregates also showed a similar loss of NE integrity (Fig. 5 G). While the increased amount of neurons with loss of NE integrity likely reflects impaired NE repair, it is important to note that reduced reimport rates of mCherry-NLS post-rupture might also contribute to this phenotype. Nonetheless, these findings do demonstrate that nuclear aggregate-induced NE ruptures and long-term loss of NE integrity also occur in the cell type primarily affected in HD.

Taken together, we have shown that while cytosolic and nuclear polyQ aggregates can both deform the nuclear lamina, only nuclear aggregates frequently induce NE ruptures (Fig. 5 D). Similar ruptures have been found in laminopathies and migrating cancer cells (Denais et al., 2016; Earle et al., 2020), where they were reported to trigger nucleocytoplasmic mixing and DNA damage (Denais et al., 2016; Earle et al., 2020; Nader et al., 2021; Xia et al., 2018). Crucially, while transient NE ruptures are often insufficient to trigger the accumulation of endogenous cGAS and subsequent STING activation (Guey et al., 2020; Hatch 2018), the prolonged loss of NE barrier function found in the presence of polyQ aggregates (Fig. 1, H–I and Fig. 5, C–F) could lead to a gradual buildup of detrimental effects, including nucleocytoplasmic mixing, transcriptional deregulation, and even DNA damage, all features that are found in many HD models (Gasset-Rosa et al., 2017; Grima et al., 2017; Sugars and Rubinsztein 2003). Because several other neurodegenerative diseases also show intranuclear aggregation (Al-Sarraj et al., 2011; Gomez-Deza et al., 2015; Skinner et al., 1975; Takahashi et al., 2001; Wen et al., 2014; Wils et al., 2010), alterations in nuclear morphology, and impaired nuclear barrier function (Baron et al., 2017; Chou et al., 2018; Fallini et al., 2020;

Freibaum et al., 2015; Lin et al., 2021; Ryan et al., 2022; Skinner et al., 1975; Takahashi et al., 2001; Zhang et al., 2015; Zhang et al., 2020), we speculate that nuclear aggregate-induced ruptures represent a unifying contributor to neurodegeneration that initiates a cascade of deregulated processes, culminating in degeneration and deleterious inflammation, a characteristic of most neurodegenerative diseases (Glass et al., 2010; Sharma et al., 2020).

## Materials and methods
### Cell culture and cell lines
U2OS-WT and U2OS-RFP-NLS cells were cultured in DMEM (HPSTA; Capricorn) supplemented with 10% FBS and 1% penicillin and streptomycin. RPE-1 cells were cultured in DMEM-F12 (ref. 11320033; Gibco) supplemented with 10% FBS and 1% penicillin and streptomycin. Cells were kept at 37°C and 5% $CO_2$. U2OS-WT was purchased from ATCC (ref. HTB-96). RPE-1 cells were a kind gift from Anna Akhmanova (Utrecht University, Utrecht, Netherlands). U2OS-RFP-NLS cells were a gift from Martin Hetzer (ISTA, Klosterneuburg, Austria) (Hatch and Hetzer 2016; Vargas et al., 2012) and were cultured in full DMEM supplemented with 0.5 mg/ml G418 (ref. ab144261, lot. GR 162868-8; Abcam). Cells were regularly tested for mycoplasma using MycoAlert Mycoplasma Detection Kit (Lonza).

### Constructs
The HaloTag-lamin A construct was generated by first inserting GFP-lamin A from pBABE-puro-GFP-lamin A into pEGFP-C2 using NheI and BamHI. Then, the HaloTag fragment was generated by PCR and inserted into pEGFP-lamin A using restriction digestion with NheI and BglII to generate HaloTag-lamin A. Constructs encoding for nuclear targeted or untargeted huntingtin exon1 fragment with polyglutamine stretch (polyQ23-NLS, polyQ74-NLS and polyQ74) were generated previously (Hageman et al., 2010). The mCherry-cGAS was a gift from Dennis E. Discher (University of Pennsylvania, Philadelphia, PA, USA) (Harding et al., 2017; Xia et al., 2018). For visualization of rupture sites in TREx microscopy, mCherry-cGAS was inserted into an HA-containing vector (gift from Qing Zhong; #280274; Addgene plasmid) (Fan et al., 2010) generating HA-mCherry-cGAS. emiRFP670-cGAS was generated by inserting cGAS PCR product into an emiRFP670-containing vector using Gibson assembly. pBABE-puro-GFP-lamin A was a gift from Tom Misteli (#17662; Addgene plasmid) (Scaffidi and Misteli 2008).

### Plasmids and transfection
U2OS WT, RPE-1, and U2OS-RFP-NLS were plated on 23-mm or 18-mm diameter coverslips 1–2 days before transfection. 1 day before live-cell imaging or fixation, cells were transfected using FuGENE6 (Promega) at a ratio of 3 µl transfection reagent per 1 µg DNA. Cells were either fixed or used for live cell imaging 1 day after transfection.

### Rat hippocampal neuron culture and transfection
Primary hippocampal cultures were isolated from embryonic rat brains (day 18) as described previously (Kapitein et al. 2010). In

brief, cells were plated on coverslips coated with laminin (2 µg ml⁻¹) and poly-*L*-lysine (30 µg ml⁻¹). Cultures were grown in full Neurobasal medium (NB, ref. 21103049; Gibco) with B27 (ref. 17504044; Gibco), 0.5 mM glutamine, 12.5 µM glutamate, and penicillin/streptomycin. Neurons were cultured at 37°C in 5% $CO_2$ for 9 days prior to transfection. Per well, transfection was performed using a transfection mix containing 1.8 µg DNA and 3.3 µl lipofectamine 2000 (ref. 11668019; Invitrogen) in 200 µl NB. The transfection mix was thoroughly mixed and incubated for 30 min. Neurons were transfected by adding the transfection mix to neurons in NB with 0.5 mM glutamine for 60 min. During transfection, neurons were kept at 37°C in 5% $CO_2$. After transfection, neurons were washed with NB and returned to full NB. Neurons were imaged, or fixed using 4% paraformaldehyde (PFA) with sucrose, 48 h after transfection.

### Live-cell and fluorescence microscopy

For live-cell imaging of U2OS and U2OS-RFP-NLS, cells were imaged on a Nikon Eclipse Ti equipped with an incubator chamber (INUG2-ZILCS0H2; Tokai Hit) on a motorized stage (ASI). Illumination was performed using a CoolLED pE4000 (CoolLED) LED device and ET-EGFP (49002; Chroma), ET-mCherry (49008; Chroma), ET-CY5 (49006; Chroma), and ET-CY5.5 (49022; Chroma) filters. All images were acquired with a Coolsnap HQ2 CCD camera (Photometrics). The microscope was controled using µManager software (Edelstein et al., 2014). Cells were imaged in a metal imaging chamber (ref. A7816; Invitrogen) that was sealed by placing a coverslip on top to prevent medium evaporation.

For long-term time-lapse imaging used for quantification of nuclear envelope rupture, blebbing, and recovery in U2OS-RFP-NLS, cells were imaged using a 20x dry objective (Plan Apo. NA 0.75; Nikon) every 5 min for 8 h. All other live cell imaging was done using a 40× oil immersion objective (Plan Fluor, NA 1.3; Nikon). For Halotag-lamin A and mCherry- or emiRFP670-GAS imaging, cells were imaged every 2–3 min for 2–6 h. For experiments with HaloTag-lamin A, cells were incubated with JF 646 (ref. GA112A, lot. 0000486504; Janelia Fluor) for 30 min prior to imaging. For live-cell quantification of NE ruptures, cells were treated with 2 mM thymidine (ref. 6060-5GM, lot. D0017544; Calbiochem) to prevent NE breakdown caused by mitosis. For the live-cell viability assay, cells were imaged with minimal excitation every 5 min for ~8 h. Mitoview 633 (ref. 70055-T, lot. 10M031-1227011; Biotium) was added 15 min before the start of imaging at 50 nM final concentration.

For single time point live-cell imaging of hippocampal neurons, we used a Nikon Eclipse Ti Microscope equipped with a CSU-X1-A1 confocal head (Yokagawa). Cells were imaged on an ASI motorized stage (MS-200-XYZ) equipped with a Piezo top plate and incubation chamber (INUBG2E-ZILCS; Tokai Hit) to capture z-stack images (0.5 µm z-spacing) using a Plan Fluor 40× N.A. 1.30 oil immersion objective (Nikon). Excitation was performed using 491 and 561 nm lasers (Cobolt Jive and Cobolt Calypso, respectively). We used ET-mCherry (49008; Chroma) and ET-GFP (49002; Chroma) filter cubes. The microscope was controled using Metamorph 7.7 software (Molecular Devices).

Fixed-cell immunofluorescence microscopy images used for scoring of lamin B1 phenotype, cGAS intranuclear accumulation or scoring of BAF, emerin, LAP2b and CHMP4B foci were taken on a Nikon Eclipse Ni-U microscope with a 60× (plan Apo Lambda, N.A. 1.40; Nikon) or 40× (Plan Fluor 40× N.A. 1.30) oil immersion objective and equipped with ET-EGFP (49002; Chroma) and ET-mCherry (49008) filters. All other fixed cell imaging was performed using a point-scanning confocal Zeiss AiryScan LSM880 microscope using a 63× immersion objective (Plan-Apochromat, 1.2 NA), controled by Zen Black software.

The contribution of actin contractility to rupture induction was probed by transfecting cells with polyQ74-NLS and mCherry-cGAS. Cells were treated with blebbistatin (final concentration of 50 µM; Sigma-Aldrich) or DMSO, and thymidine (to block mitosis as previously described) ~8 h after transfection. After ~22 h, cells were fixed and used for imaging.

Images of the expanded cells were acquired using a Leica TCS SP8 STED 3X microscope equipped with an HC PL APO ×86/1.20 W motCORR STED (15506333; Leica) water objective controled using Leica Application Suite X.

### Antibodies and reagents

For immunofluorescence labeling, the following antibodies were used: anti-lamin A/C (ref. sc-7292, lot. L1919; Santa Cruz), anti-lamin B1 (ref. ab160848, lot. GR3417466-1; Abcam), anti-cGAS (ref. 15102, lot. 4; Cell Signaling Technology), anti-GFP (ref. 598, lot. 081; MBL-Sanbio), anti-HA (ref. SC-57592, lot. L2310; Santa Cruz), anti-GFP (ref. GFP-1010, lot. GFP3717982; Aves Labs), anti-BAF (ref. ab129184, lot. GR3403496-7; Abcam), anti-CHMP4B (ref. 13683-1-AP, lot. 00110324; Thermo Fisher Scientific), anti-Emerin (8F5A8) (ref. ab204987, lot. GR3350015-7; Abcam) and anti-LAP2beta (ref. 611000, lot. 2196090; Thermo Fisher Scientific). Secondary antibody labeling was done using goat anti-rabbit 488 (ref. A11034, lot. 2256692 and 2286890; Thermo Fisher Scientific), goat anti-rabbit 568 (ref. A11036, lot. 2045347; Thermo Fisher Scientific), goat anti-mouse 568 (ref. A11031, lot. 2124366; Thermo Fisher Scientific), goat anti-mouse 594 (ref. A11032, lot. 2069816 and 2397936; Thermo Fisher Scientific), goat anti-chicken 488 (ref. SA5-10070, lot. VI3075603; Thermo Fisher Scientific), and goat anti-rabbit 594 (ref. A11037, lot. 2160431; Thermo Fisher Scientific).

### Immunofluorescence

For immunofluorescence labeling for wide-field or confocal fixed-cell imaging, cells were first fixed using prewarmed 4% PFA. Cells were washed with 1X PBS (Lonza), permeabilized using 0.2% Triton-X100, and blocked using blocking solution (3% BSA in 1X PBS) for 1 h at RT. Cells were incubated with primary antibody (1:500 in blocking solution) at 4°C overnight. Cells were subsequently washed in 1X PBS and incubated with the appropriate secondary antibodies (1:500 in blocking solution) for 1 h at RT. Cells were dried and mounted using Prolong Diamond Antifade Mountant (ref. P36965; Invitrogen).

### Expansion microscopy
#### Fixation and pre-extraction
For 10-fold robust expansion (TREx) microscopy, we adapted the protocol described previously (Damstra et al., 2022). For mCLING staining, cells were fixed using prewarmed 4% PFA, 4%

sucrose (wt/vol), and 0.1% glutaraldehyde. Coverslips were incubated with 10 μM mCLING (710 006AT1; Synaptic Systems) for 4–6 h at 37°C and subsequently incubated overnight at RT. The incubated samples were postfixed with prewarmed 4% PFA and 0.1% glutaraldehyde. Cells that were not stained using mCLING were only fixed with prewarmed 4% PFA. For total protein labeling, we incubated neurons with Atto 643 NHS ester (AD 643-35; Atto-Tec) at 30 mg/ml (in 1X PBS). To reduce background obscuring smaller cytosolic polyQ74 aggregates in U2OS cells, we pre-extracted cells by incubation with 1 ml of 0.15% (vol/vol) prewarmed Triton X-100 in 1X PBS for 1 min prior to fixation.

### Gelation and expansion
The following monomer solution was prepared on ice: 1.085 M sodium acrylate (408220; Sigma-Aldrich), 2.664 M acrylamide (A4058; Sigma-Aldrich), and 0.009% (vol/vol) N,N′-methylenebisacrylamide (M1533; Sigma-Aldrich) in 1x PBS. The polymerization reaction was started by the addition of 1.5% (vol/vol) tetramethylethylenediamine (TEMED) and 1.5% (vol/vol) ammonium persulfate (APS). The monomer solution was vortexed and 170 μl (per coverslip) was pipetted into a silicon gelation chamber attached to a parafilm-covered glass slide. The coverslip containing stained cells was blotted onto the monomer solution. The gelation chambers were incubated at 37°C for 1 h. Subsequently, gels were digested in 2 ml of digestion mix for 4 h at 37°C and expanded up to 10x using MilliQ.

### Data processing and 3D-rendering
Prior to image analysis, all TREx images except mCLING channels were deconvolved with Huygens Professional version 21.04 (Scientific Volume Imaging, https://svi.nl) using the CMLE algorithm with 4 SNR and 20 iterations. mCLING channels were blurred using Gaussian Blur 3D with 0.8 Sigma-Aldrich (both X, Y, and Z) and subsequently used to generate a rolling average using Running Z projector (https://valelab4.ucsf.edu/~nstuurman/IJplugins/Running_ZProjector.html) with a running average size of three slices per slice. For 3D volume rendering Arivis Vision4D version 3.5.0 was used. PolyQ74-NLS filaments were processed by normalizing the intensity (method: simple), detected using a "random forest" machine learning classifier (https://ukoethe.github.io/vigra/), segmented by intensity thresholding, and filtered by size (>0.001 μm³) and manual deletion. The lamin B1 network was processed by normalizing the intensity (method: simple) and masking the external nuclear signal using a manually generated mask.

### Quantifications
Cells were manually scored for quantification of at least one nuclear envelope rupture or nuclear blebbing event during 8-h imaging. Cells that divided, died, or migrated out of the field of view of the camera were excluded from the analysis. Cells that did not have nuclear RFP-NLS enrichment at the start of imaging were also excluded from the analysis. The fraction of cells that recovered nuclear enrichment of RFP-NLS intensity was determined by manually scoring recovery in all cells that showed a nuclear rupture event. Cells that ruptured in the last hour of imaging (7–8 h after start of imaging) were excluded from the

analysis. For quantification of recovery dynamics after nuclear rupture, RFP-NLS intensity ratios (Nuclear/Cytoplasmic) were normalized to the average intensity ratio of three frames before rupture. Recovery half-time was calculated by normalizing individual recovery traces and setting the lowest nuclear enrichment value as $t = 0$. Time to 50% recovery was determined to be the point when each trace first recovered up to 50% of the prerupture enrichment.

Quantification of cellular viability was done by identifying unhealed ruptures in cells based on RFP-NLS loss from the nucleus. Although Mitoview 633 shows some fluorescence in the RFP-channel, ruptures could still be identified clearly in these cells (Fig. S1, K and M). These cells were then scored as having maintained or lost ΔΨm at various points during imaging. Cells that already lost Mitoview 633 signal at the start of imaging were not included in the analysis.

The effect of polyQ aggregate presence in the nucleus or cytosol on the lamin B1, BAF, CHMP4B, emerin, or LAP2b was determined by manually scoring endogenous protein localization in control U2OSWT cells or cells expressing polyQ23-NLS, polyQ74-NLS, or polyQ74. For lamin B1, the percentage of cells showing either disruption (absence of lamin B1 signal at nuclear rim) or, unifocal deformation (single deformation of lamin B1) was quantified. For BAF, CHMP4B, emerin, and LAP2 stainings the amount of cells with at least one nuclear focus was scored. Cells that appeared dead, rounded up, or dividing were excluded from the analysis.

Quantification of nuclear rupture frequency in neurons was done by manually scoring intranuclear cGAS accumulation in cells that were transfected (control and polyQ23-NLS) or showed aggregates (polyQ74-NLS and polyQ74). Neuron viability was assessed by neuronal morphology in the BFP-fill signal. Neurons that were dead or that did not express sufficient BFP-fill to visualize neuronal processes were excluded from the analysis. To investigate prolonged disruption of nuclear barrier function in neurons, we calculated the mCherry-NLS ratios ($I_{nuclear}$ mCh/$I_{cytosolic}$ mCh). Background-corrected intensity measurements were performed after maximum intensity projections. To determine the fraction of nuclei without enrichment, we calculated the amount of cells with mCherry-NLS ratios below 1.5.

### Statistical analysis
Prism9 (GraphPad) was used for generating all graphs and statistical analyses. Differences in nuclear rupture frequency, blebbing frequency, and recovery frequency were tested for statistical significance using a Fischer's exact test with the null hypothesis that proportions were equal in all groups. Differences in percentages of hippocampal neurons showing intranuclear cGAS accumulation, and of neurons with low nuclear enrichment of mCherry-NLS, were also tested for statistical significance using Fischer's exact test. Similarly, differences in the frequency of foci formation of BAF, CHMP4B, emerin, and LAP2b were tested for significant using Fischer's exact test. Lamin B1 phenotypes were tested for significant differences using an ordinary one-way ANOVA and Tukey's post hoc test or Fischer's exact test. All Fischer's exact tests were corrected for multiple testing using Bonferroni correction.

Recovery dynamics after NE rupture were measured by first calculating traces of nuclear enrichment (Nuclear enrichment = $I_{nuclear\ RFP}/I_{cytosolic\ RFP}$) per cell normalized to prerupture enrichment values. Average nuclear enrichment was calculated by aligning normalized enrichment values to the time of NE rupture.

To test whether mCherry-NLS ratios were significantly different for neurons expressing polyQ23-NLS and polyQ74-NLS, we used an unpaired Student's $t$ test. The distribution of the data used for parametric tests was tested for normality using a Shapiro–Wilk test. To determine whether the cumulative frequency distributions of mCherry-NLS ratios were different between these groups, we used a Mann–Whitney test.

All quantifications of intensity values in U2OSWT and U2OS-RFP-NLS cells expressing polyQ74, polyQ74-NLS, mCherry-cGAS, or HaloTag-lamin A were performed on raw, unprocessed, images. Values were background-corrected before plotting.

### Data processing

All image processing of non-expanded samples was performed using FIJI (Schindelin et al., 2012). Multiple images of the intense signal of polyQ aggregates or cGAS were gamma corrected ($\gamma$ = 0.50–0.75) to increase the visibility of the signal around intense cores (see figure legend). Final figure panels were prepared in Adobe Illustrator.

### Online supplemental material

Fig. S1 shows additional data on NE blebbing, deformation, and rupture in cells expressing polyQ aggregates and shows maintained cell viability after prolonged loss of NE integrity. Related to Fig. 1. Fig. S2 shows the localization of cGAS and lamin A at aggregate-induced rupture sites. Related to Fig. 2. Fig. S3 contains additional expansion data revealing polyQ aggregate ultrastructure and interactions with the NE and lamina. Related to Figs. 4 and 5. Video 1 (related to Fig. 1 A) shows NE rupture in a U2OS-RFP-NLS cell with nuclear aggregates. Video 2 (related to Fig. 1 H) shows prolonged loss of NE integrity after NE rupture in a U2OS-RFP-NLS cell expressing nuclear polyQ aggregates. Video 3 (related to Fig. 2 B) shows accumulation of mCherry-cGAS around a nuclear polyQ aggregate in a U2OS cell. Video 4 (related to Fig. 2 F) shows scar formation of Halotag-LaminA upon NE rupture in a U2OS-RFP-NLS cell expressing nuclear polyQ aggregates. Video 5 (related to Fig. 4 A; right panel) shows a volumetric render of nuclear polyQ aggregates in an expanded U2OS cell.

### Data availability

All data underlying the figures are available via FigShare: https://figshare.com/collections/Nuclear_poly-glutamine_aggregates_rupture_the_nuclear_envelope_and_hinder_its_repair/7345940. Plasmids are available from L.C. Kapitein upon request.

## Acknowledgments

We thank Martin Hetzer (ISTA, Klosterneuburg, Austria) for the gift of U2OS-RFP-NLS cells, Fulvio Reggiori, and Anna Akhmanova for discussions and advice, Anne Janssen for early observations of nuclear ruptures in cells with aggregates, Hugo Damstra for help with expansion microscopy, and Klara Jansen for advice regarding primary neuron culture.

This research was supported by the European Research Council (ERC Consolidator Grant 819219 to L.C. Kapitein) and the Dutch Research Council (NWO, ZonMW 91217002 to H.H. Kampinga and L.C. Kapitein).

Author contributions: G. Korsten: Conceptualization, Data curation, Formal analysis, Investigation, Methodology, Project administration, Validation, Visualization, Writing—original draft, Writing—review & editing, M. Osinga: Formal analysis, Investigation, Visualization, Writing—review & editing, R.A. Pelle: Formal analysis, Investigation, Visualization, Writing—review & editing, A.K. Serweta: Investigation, B. Hoogenberg: Investigation, Methodology, Visualization, H.H. Kampinga: Conceptualization, Funding acquisition, Methodology, Resources, Supervision, Writing—review & editing, L.C. Kapitein: Conceptualization, Funding acquisition, Project administration, Supervision, Writing—original draft, Writing—review & editing.

Disclosures: The authors declare no competing interests exist.

Submitted: 28 July 2023

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

# Supplemental material

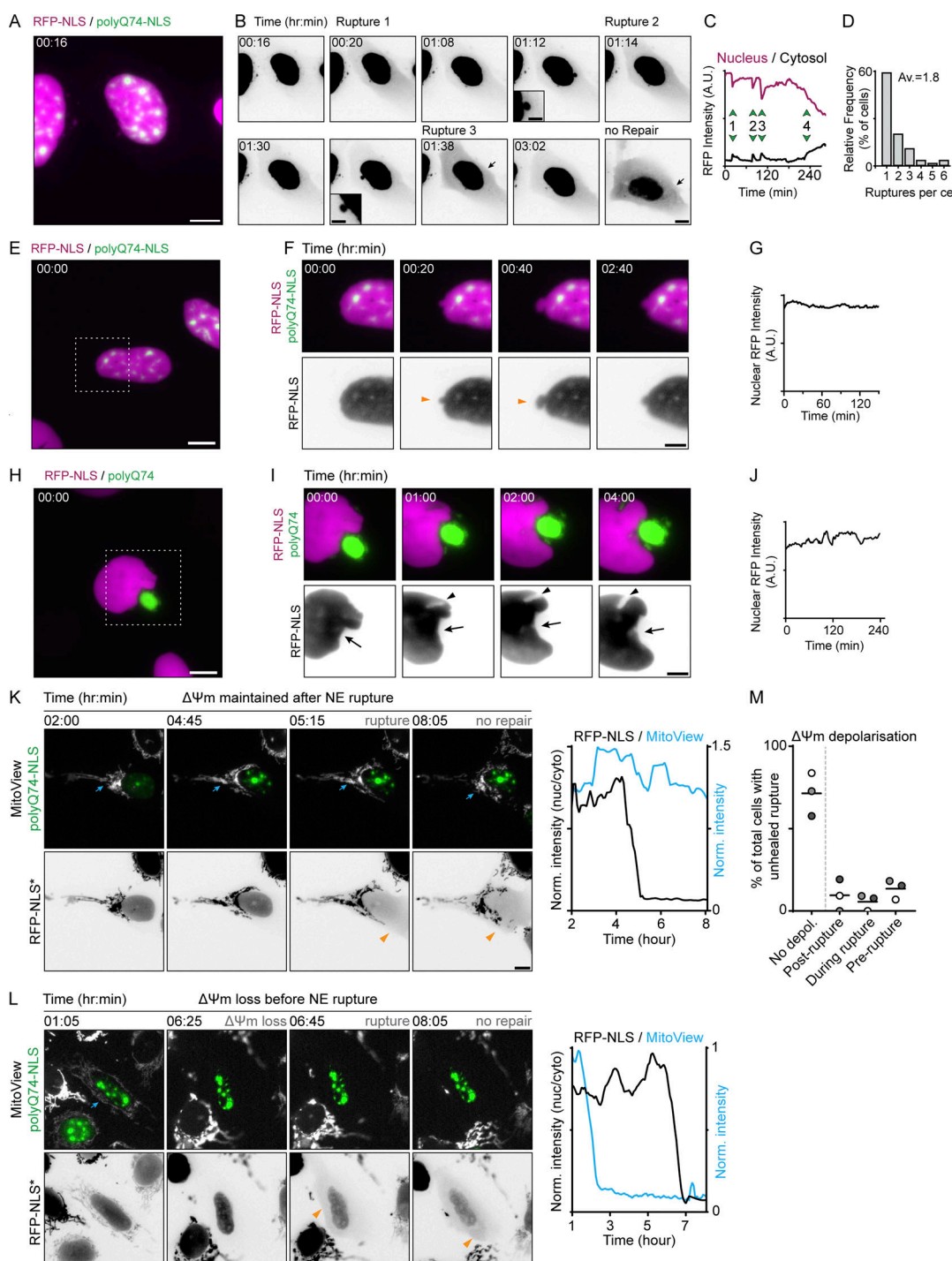

Figure S1. **Studying NE rupture and viability in cells expressing polyQ aggregates. (A)** U2OS-RFP-NLS (magenta) cells expressing polyQ74-NLS aggregates (green). **(B)** Timelapse of RFP-NLS signal (inverted grayscale) of cell shown in A showing multiple blebbing events (black arrowheads) and NE rupture evidenced by cytosolic leaking of RFP-NLS signal (black arrow). **(C)** Graph showing nuclear (magenta curve) and cytosolic (black curve) RFP-intensity, indicating multiple instances of changes in nuclear and cytosolic RFP-NLS signal (green arrows 1, 2, and 3). After the last nuclear rupture (green arrow 4), nuclear enrichment does not restore. **(D)** Percentages of cells showing various amounts of nuclear ruptures during 8-h imaging. **(E and F)** Time-lapse images of representative U2OS-RFP-NLS cells (magenta and inverted grayscale) and polyQ74-NLS (green) showing blebbing event (orange arrowheads) without NE rupture. **(G)** Graph of nuclear RFP-intensity of timelapse in F, showing no loss of RFP-NLS signal during blebbing event. **(H and I)** Time-lapse images of representative U2OS-RFP-NLS cells expressing cytosolic polyQ74 aggregates (green), showing a characteristic bean shaped nucleus (black arrow) and NE deformation near a smaller cytoplasmic aggregate (black arrowhead). **(J)** Nuclear RFP-intensity of timelapse in I showing no loss of RFP-NLS signal during imaging. **(K and L)** Representative images of live-cell imaging of U2OS-RFP-NLS cells expressing polyQ74-NLS (green) that were labeled using Mitoview (grayscale) showing maintenance (K) or loss (L) of mitochondrial membrane potential in cells with unhealed NE ruptures. **(M)** Quantification of the fraction of cells with unhealed NE ruptures showing maintenance, or loss, of mitochondrial membrane potential (n = 80; N = 3). All scale bars indicate 10 μm, except for F and I and zooms of B (5 μm).

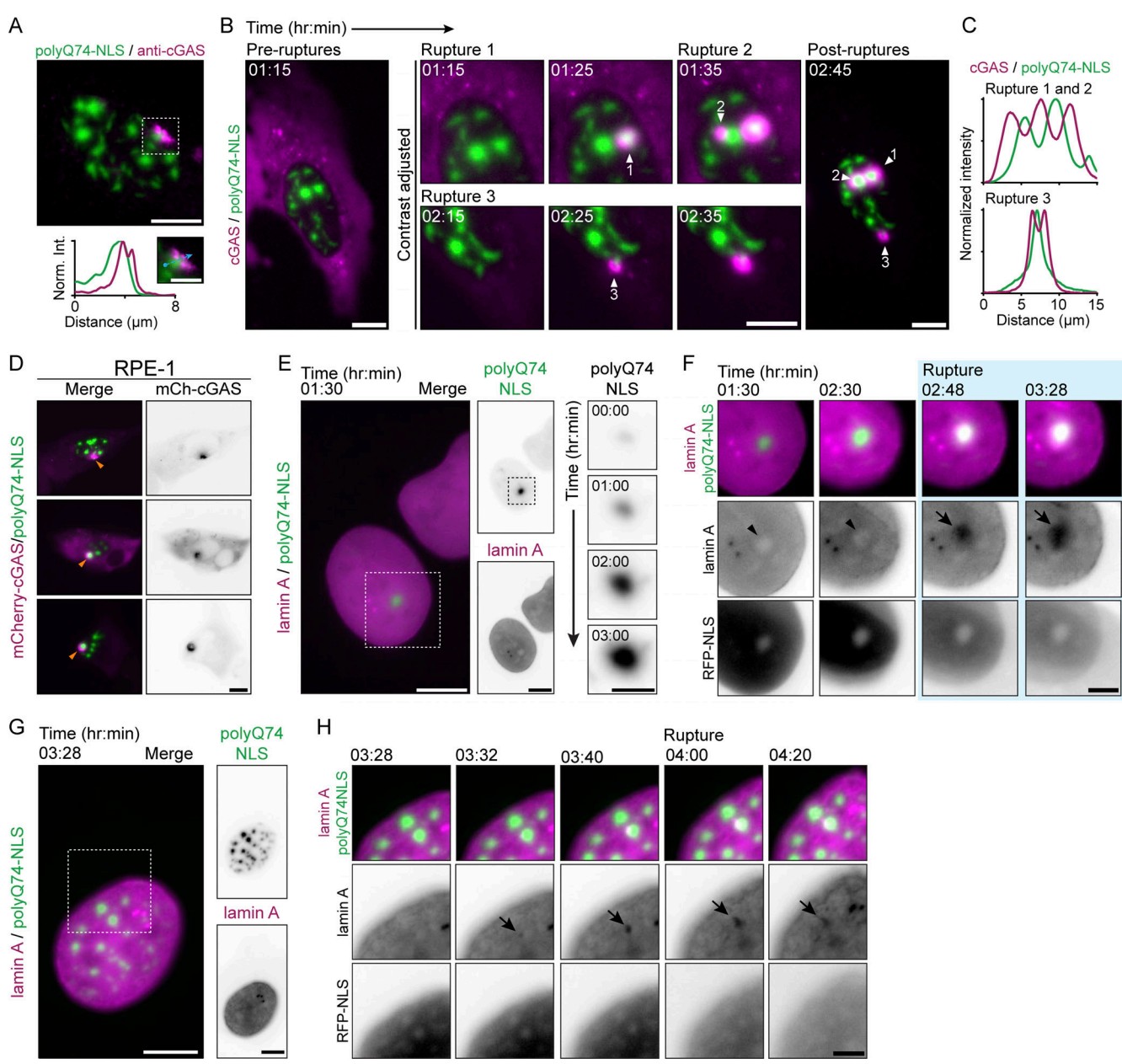

Figure S2. **cGAS and lamin A localization at aggregate induced rupture sites. (A)** Representative image of U2OSWT cells expressing polyQ74-NLS (green), immunostained for endogenous cGAS (magenta). Profile plot of line scan in inset, showing localization of endogenous cGAS around a nuclear aggregate. **(B)** Representative stills from live-cell imaging of U2OSWT cells expressing polyQ74-NLS (green) and mCherry-cGAS (magenta). Time-lapse zooms show multiple ruptures (Rupture 1, 2 and 3) occurring at different aggregates. **(C)** Graphs of intensity profiles across rupture sites shown in B, indicating mCherry-cGAS accumulation around individual aggregates. **(D)** RPE-1 cells expressing polyQ74-NLS aggregates and mCherry-cGAS. Orange arrows indicate nuclear cGAS accumulation at polyQ aggregates. **(E and G)** Representative U2OS-RFP-NLS cells expressing polyQ74-NLS (green) and HaloTag-lamin A (magenta). Zooms of E showing aggregate growth during imaging. **(F)** Timelapse of zooms of cell shown in E. Stills show HaloTag-lamin A scar formation upon rupture (black arrows) at nuclear polyQ74-NLS aggregates. **(H)** Time-lapse images of the cell shown in G with minor HaloTag-lamin A accumulation (black arrow) at a nuclear aggregate. PolyQ and cGAS signal was gamma adjusted (γ = 0.75). Scale bars represent 10 μm (A, B, D, E, and G) and 5 μm (A_zoom, E_zoom, F, and H).

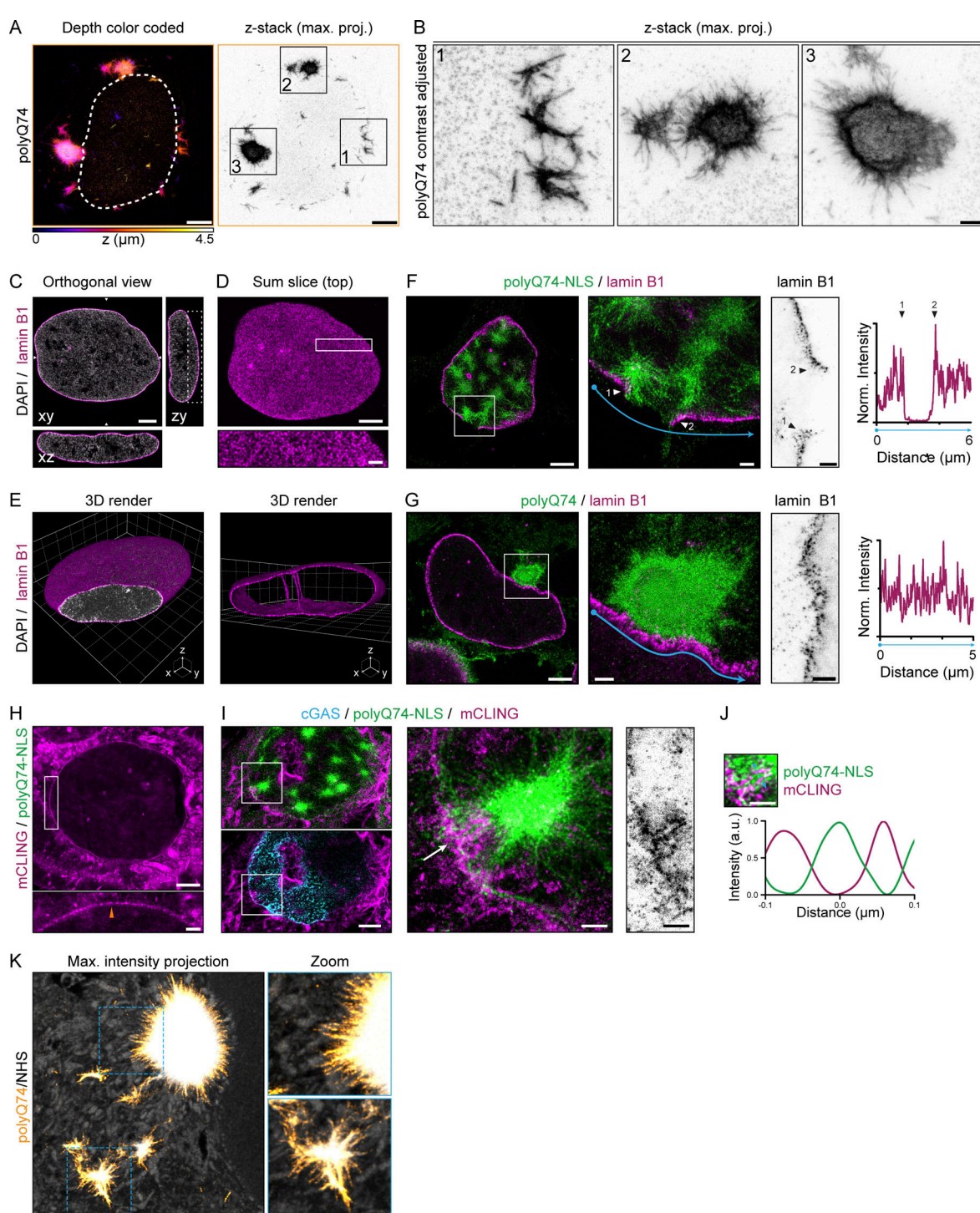

Figure S3. **TREx microscopy reveals lamin B1 network and NE deformations induced by nuclear polyQ aggregates. (A)** Depth color coded (left panel) and maximum intensity projection (right panel) of a representative expanded (TREx) U2OS cell with cytosolic polyQ74 aggregates. **(B)** Zooms of various cytosolic aggregates shown in A. **(C–E)** Images of representative expanded U2OS cell immunolabeled for lamin B1 (magenta) and counterstained with DAPI (gray). **(C)** Orthogonal view showing discrete localization of lamin B1 at the nuclear periphery. **(D)** Sum projection of the top 1.5 µm of the nucleus shown in C, showing a zoom of the lamin B1 meshwork. **(E)** 3D render of the whole lamin B1 meshwork of cell shown in C. Cropped render of the middle of the nucleus reveals nucleoplasmic reticulum present in U2OS cells. **(F and G)** Representative images of expanded U2OS cells expressing polyQ74-NLS (F) or polyQ74 (G) stained for lamin B1. Lamin B1 intensity profile indicates disrupted (F) or intact (G) lamin B1 meshwork at aggregate. **(H)** Control U2OS cell stained with mCLING (magenta) showing a distinct NE (orange arrow). **(I)** Images of expanded U2OS cell expressing polyQ74-NLS and HA-mCherry-cGAS (cyan), showing NE deformation and invagination around a nuclear aggregate, at a NE rupture location marked by cGAS accumulation. **(J)** Intensity profile of NE rupture site shown in G showing mCLING stained NE accumulation around a polyQ fibril. **(K)** Representative images of expanded neurons expressing cytosolic polyQ74 aggregates (orange) labeled using a total protein stain (NHS; grayscale). Scale bars indicate 2.5 µm for overviews and 0.5 µm for zooms (A–D, F–J, and K). Voxel sizes are 2 µm (grids in E).

Video 1. **NE blebbing and rupture induced by nuclear polyQ aggregates.** Time-lapse movie (8 frames/s) of representative U2OS-RFP-NLS cell expressing RFP-NLS (magenta) and polyQ74-NLS (green) aggregates (stills are shown in Fig. 1 C). Cell shows multiple NE blebbing events followed by a transient NE rupture. Cells were imaged every 2 min using epifluorescence microscopy. Arrowheads indicate blebbing and rupture events. Scale bar indicates 10 μm.

Video 2. **NE blebbing and unhealed NE rupture induced by nuclear polyQ aggregates.** Time-lapse movie (8 frames/s) of representative U2OS-RFP-NLS cell expressing RFP-NLS (magenta) and polyQ74-NLS (green) aggregates (stills are shown in Fig. 1 H). The cell shown undergoes NE rupture without subsequent restoration of nuclear RFP-NLS enrichment. Cells were imaged every 2 min using epifluorescence microscopy. Arrowhead indicates blebbing and rupture event. Scale bar indicates 10 μm.

Video 3. **mCherry-cGAS rapidly accumulates close to polyQ aggregates after NE rupture.** Time-lapse movie (8 frames/s) of representative U2OS-WT cell expressing mCherry-cGAS (magenta) and polyQ74-NLS (green) aggregates (stills are shown in Fig. 2 B). The movie shows rapid accumulation of mCherry-cGAS close to a nuclear polyQ aggregate following NE rupture. The movie includes an additional panel showing contrast adjusted mCherry-cGAS signal to aid visibility. Cells were imaged every 5 min using epifluorescence microscopy. Arrowhead indicates rupture event. Scale bar indicates 10 μm.

Video 4. **LaminA depletion and scar formation during polyQ aggregates induced NE rupture.** Time-lapse movie (8 frames/s) of representative U2OS-RFP-NLS cell expressing RFP-NLS (inverted grayscale), Halotag-LaminA (magenta) and polyQ74-NLS (green) aggregates (stills are shown in Fig. 2, F and G). The movie shows Halotag-LaminA depletion and scar formation around a nuclear aggregate before and after NE rupture, respectively. Cells were imaged every 2 min using epifluorescence microscopy. Arrowhead indicates rupture event. Scale bar indicates 10 μm.

Video 5. **TREx reveals nuclear polyQ aggregate ultrastructure in a whole nucleus.** Volumetric render of nuclear polyQ74-NLS aggregates (green) in a representative expanded U2OS-WT cell (stills are shown in Fig. 4 A; right panel). Expanded cell was imaged after 10-fold expansion using confocal microscope microscopy. The scalebar is shown as an imbedded voxel grid of 2 by 2 μm.

