## [Peer Review File · The Journal of Cell Biology]

Nuclear poly-glutamine aggregates rupture the nuclear envelope and hinder its repair

Giel Korsten, Miriam Osinga, Robin Pelle, Albert Serweta, Baukje Hoogenberg, Harm Kampinga, and Lukas Kapitein

Corresponding Author(s): Lukas Kapitein, Utrecht University

Review Timeline:

Submission Date:	2023-07-28
Editorial Decision:	2023-09-07
Revision Received:	2024-04-08
Editorial Decision:	2024-06-22
Revision Received:	2024-07-12

Monitoring Editor: Martin Hetzer

Scientific Editor: Dan Simon

Transaction Report:

DOI: <https://doi.org/10.1083/jcb.202307142>

September 7, 2023

Re: JCB manuscript #202307142

Prof. Lukas Kapitein
Utrecht University
Padualaan 8
Utrecht 3533 CH
Netherlands

Dear Prof. Kapitein,

Thank you for submitting your manuscript entitled "Nuclear poly-glutamine aggregates rupture the nuclear envelope and hinder its repair" to Journal of Cell Biology. The manuscript has now been assessed by expert reviewers, whose reports are appended below. Unfortunately, after an assessment of the reviewer feedback, our editorial decision is against publication in JCB.

You will see that the reviewers feel that the work is well done and presents some intriguing data. However, they also expressed low enthusiasm for the suitability of the study to JCB due to concerns about the relevance of observations made in non-neuronal U2OS cells and from expression of artificial polyQ74-NLS in neurons.

Unfortunately we do not have the level of reviewer support that we would need to proceed further with the paper. We do realize that significant further work and expansion might convincingly address some of these issues, but we are hesitant to encourage you to work towards the aim of further consideration at JCB. The level of reviewer criticism makes it impossible for us to guarantee that we will be able to invite resubmission, even after revision. Therefore, it does seem that it will be best for you to consider another journal for this work. Our journal office can transfer your reviewer comments to another journal upon request.

We are sorry our decision is not more positive, but hope that you find the reviews constructive. Of course, this decision does not imply any lack of interest in your work and we look forward to future submissions from your lab.

Thank you for your interest in Journal of Cell Biology.

Sincerely,

Martin Hetzer, PhD
Monitoring Editor
Journal of Cell Biology

Dan Simon, PhD
Scientific Editor
Journal of Cell Biology

Reviewer #1 (Comments to the Authors (Required)):

In this manuscript Korsten et al study the effects of nuclear poly-glutamine aggregates on nuclear compartmentalization. They demonstrate that expression of an expanded polyQ huntingtin fragment (polyQ74), a model for Huntington's disease, causes nuclear rupture in primary neurons and in U2OS cells. They further describe the effects of nuclear polyQ aggregates on nucleus shape, nuclear lamina organization, and nuclear membrane resealing after rupture.

The strongest point of this study is the observation that polyQ74 induces nucleus rupture in neurons. This represents a new mechanistic defect in Huntington's cells that could underlie important pathologies. However, the experiments in U2OS cells examining whether polyQ74 directly causes rupture, how it impacts repair, and its effects on lamina organization are very descriptive and leave open the question of whether polyQ74 acts directly on the nuclear membrane during interphase to cause increased instability or indirectly through increased apoptosis or mitotic defects. Importantly, it is also not clear that the observations in U2OS cells are directly relevant to the situation in neurons. Overall the experiments are done well, the data presentation is easy to read, the statistics used are perfect (which is very rare for categorical data!), and the writing is clear and compelling. However, based on the general nature of the data and the unclear relevance of the U2OS model due to its inherent defects in nuclear envelope structure, I think this paper would be better suited for a different journal.

Major issues

1. The methods indicate that the major experiments in figure 1, quantifying nuclear rupture frequency in U2OS cells, and figures 2 and 3, examining lamin organization, were performed during S-phase arrest. This increases both nuclear lamina disruptions

and nuclear rupture frequency, potentially masking or enabling the phenotypes associated with polyQ74-NLS expression. In addition, cancer cells have known defects in nuclear envelope structure that make them a poor model for processes occurring in normal cells not under extensive mechanical force (e.g. squeezing through small pores). To understand the effect of polyQ-NLS on nuclear rupture and nuclear lamina organization in neurons the ideal platform would be a non-transformed cell line with minimal disruptions of the nuclear envelope.

2. A direct function of polyQ74-NLS in rupturing the membrane through induced curvature is not completely clear from the presented data. Although the images of polyQ74-NLS induced membrane curvature in Figure 3 are fantastic, the data in figure 1F and the figure 1 videos suggest that most of the ruptures in these cells occur in membrane blebs from which aggregates are excluded. In addition, it is likely that polyQ74 aggregates can only affect membrane curvature at lamina gaps. Thus the biological significance of the ability of polyQ74 aggregates to induce membrane curvature is unclear.

Expression of polyQ74-NLS is correlated with increased nuclear lamina disruption, but it is unclear how direct this is. One likely explanation is that the protein interferes with nuclear envelope assembly during mitosis. This could be determined by live-cell imaging or more clearly by inducing expression in interphase only. If this is mainly a mitotic phenotype in U2OS cells, it is unclear how the data in this model relates to the situation in neurons. From the videos the nuclei also appear very stiff, as if lamin A levels or chromatin organization were severely altered, which could also drive rupture indirectly.

3. A direct function of polyQ74-NLS in preventing membrane repair is not clear from the presented data. This argument rests on the observation that a proportion of ruptured nuclei with polyQ74-NLS do not repair and that polyQ-aggregates are present at rupture sites and do not co-localize with lamin A. An alternative hypothesis is that nuclei with polyQ74-NLS aggregates are more likely to undergo apoptosis, which causes persistent nuclear rupture prior to cell death. Analysis of the timing between rupture and apoptosis in polyQ74 cells could be informative, as cells can persist with ruptured nuclei (caused by BAF depletion) for many hours prior to death, as well as IF analysis of apoptotic marks, such as lamin phosphorylation in cells with ruptured nuclei. With regards to the potential mechanism, their evidence that polyQ74-NLS inhibits membrane recruitment is unclear. Both lamin A and cGAS bind chromatin (indirectly, in lamin A's case), not polyQ, and therefore their exclusion from aggregates is not unexpected. However, the images in Figure 2A, D, and E don't show the depletion depicted in the line graphs, making the biological significance of this observation unclear. In addition, their data in Figure 3 show a strong accumulation of membrane at a rupture site, suggesting that membrane can be recruited. To determine whether resealing is prevented additional analysis of rupture sites by correlative EM would be required and analysis of ESCRT-III protein localization by IF.

Minor issues

1. It is not clear initially from the test what the disease-relevance of the polyQ74-NLS construct is.

2. Images of the nuclear lamins in figure 2 are compromised by the use of Prolong Diamond mounting medium. This squashes the nucleus and gives rise to an aberrant intranuclear signal. In general, immunofluorescence analysis of nuclear envelope proteins requires mounting in a glycerol based medium. This phenomenon likely doesn't affect the authors' quantification of deformed nuclei, but it can obscure nuclear lamina disruptions by compressing the signal and makes interpretation of the images challenging.

3. Figure 11: an additional graph with the half-times for the individual recovery curves would be beneficial to determine whether there are multiple repair types present in the different groups.

Reviewer #2 (Comments to the Authors (Required)):

This study by Korsten et al reveals new cellular impacts of Huntington's disease-associated polyglutamine inclusions. These aggregates are shown to cause nuclear rupture, hinder repair mechanisms and lead to dysregulation of the nuclear lamina. Through live-cell and TReX imaging, the authors track nuclear rupturing by an NLS-directed polyQ74 construct, which results in nuclear cargo escape, cGAS response, and lamin deformation. This study builds upon previous results from Baurlein et al (11) demonstrating that polyQ aggregates form fibrils that deform membranes, as well as Gasset-Rosa et al (7) and Grima et al (8) highlighting how polyQ aggregates compromise NPC function and nucleocytoplasmic transport. Significantly, the present study reports that polyQ aggregates directly initiate membrane rupture and impede repair by extension of fibrils through the rupture site. The authors also show disruptions to the nuclear lamina that are reminiscent of findings by Denais et al (19) describing membrane rupture and repair during cell migration. Overall, we find the experimental approaches and interpretations of results sound, making this manuscript a valuable contribution to further understanding the pathology of Huntington's disease. We have a few concerns that should be addressed experimentally, with the main concern centering around the artificially created polyQ74-NLS construct and whether it recapitulates the effects of disease-associated variations.

Major concerns

1. Can the authors rule out that appending the highly charged NLS changes the properties of polyQ74 that induce nuclear rupture? This construct is observed to localize within the nucleus leading to inclusions with more fibrillar morphology compared to non-NLS bearing variants. Is there any evidence (for non-NLS bearing variants) that fibrillar structures analogous to polyQ74-NLS reach/breach the nuclear envelope? Do any of the cytosolic, perinuclear inclusions similarly breach the NE from the outside? Perhaps the authors could consider using a system that involves cleaving the NLS after nuclear entry via nuclear TEV, previously used by Hartl and others. Lastly, it would be informative if the authors could test an alternative, longer (and more pathogenic) variant of huntingtin exon1.

2. How is the repair process initiated after a rupture occurs? Have the authors considered the potential involvement of the ESCRT pathway and, if so, whether disruption to this pathway (either by knockdown or expression of dominant negative factors) would hinder repair of rupture sites? Even a minimalistic subset of experiments within the chosen (short) publication format would elevate the manuscript mechanistically.
3. Can the authors rationalize/discuss why polyQ74 is nuclear excluded in U2OS cells but not in the neuronal cells?
4. For the experiments with neurons, it would be helpful if the authors could perform a Z-scan to better differentiate between nuclear and cytosolic signal. Moreover, in Fig.4B, A notable buildup of cGAS is observed in cases of P74Q overexpression. The aggregates formed under these conditions show a more compact morphology, unlike the fibril structure seen in P74Q-NLS. This leads to the question of how the compact P74Q aggregates (vs. fibrils observed in U2OS with NLS-PolyQ) contribute to nuclear envelope (NE) rupture.

Minor concerns

1. Line 23: "envelope" is misspelled as "envelop" in the one-sentence summary.
2. Is there a directionality to bleb formation comparing NLS/non-NLS variants? Comparing Fig. 1C with Fig. 4J, the nuclear bleb points outward and inward, respectively.
3. In Fig. 4J, why is cGAS present - is this immediately preceding rupture?
4. Based on approximately 20% of cells in Fig. 4B, is it technically feasible to identify polyQ74 fibrils in neurons? This would nicely support the data in U2OS cells.
5. Did the authors consider using DNA leakage as a readout? In Fig. 2A or other rupture events, the authors can attempt to stain DNA to determine whether cytoplasmic exposure of DNA occurs.
6. Did the authors normalize protein expression levels across constructs (e.g. via immunoblot).
7. Can LaminA accumulation be used as a score of rupture severity (as in citation 19) and related to polyQ74 aggregate size and/or frequency?
8. polyQ74 protein is mostly referred to as polyQ74 and on three occasions as polyQ74cyto. Either name is fine as long as defined and used consistently.
9. Fig. 3J shows a magnified subfigure of polyQ74-NLS and mCLING staining that lacks cGAS staining, despite a box indicating magnification placed around the unzoomed image of cGAS and mCLING. This magnified image should include the cGAS staining to better resolve its localization around fibrils and the nuclear deformation site.
10. Are there any instances of TREx imaging showing cGAS accumulation and/or mCLING absence at rupture sites? Fig 3J shows cGAS present as a marker for nuclear rupture, yet the image lacks other signs of rupture, such as a breach in mCLING staining and/or the protrusion of fibrils through the membrane. Because mCLING does not differentiate among membranes, the authors should clarify the nature of this inward blebbing. Showing rupture-localized cGAS under conditions such as those shown in Fig. 3F would help validate cGAS as a readout for rupture.

Point-by-point response to the reviewer comments

Reviewer #1:

In this manuscript Korsten et al study the effects of nuclear poly-glutamine aggregates on nuclear compartmentalization. They demonstrate that expression of an expanded polyQ huntingtin fragment (polyQ74), a model for Huntington's disease, causes nuclear rupture in primary neurons and in U2OS cells. They further describe the effects of nuclear polyQ aggregates on nucleus shape, nuclear lamina organization, and nuclear membrane resealing after rupture.

The strongest point of this study is the observation that polyQ74 induces nucleus rupture in neurons. This represents a new mechanistic defect in Huntington's cells that could underlie important pathologies. However, the experiments in U2OS cells examining whether polyQ74 directly causes rupture, how it impacts repair, and its effects on lamina organization are very descriptive and leave open the question of whether polyQ74 acts directly on the nuclear membrane during interphase to cause increased instability or indirectly through increased apoptosis or mitotic defects. Importantly, it is also not clear that the observations in U2OS cells are directly relevant to the situation in neurons. Overall the experiments are done well, the data presentation is easy to read, the statistics used are perfect (which is very rare for categorical data!), and the writing is clear and compelling. However, based on the general nature of the data and the unclear relevance of the U2OS model due to its inherent defects in nuclear envelope structure, I think this paper would be better suited for a different journal.

- We thank the reviewer for their positive comments and constructive feedback. To address the main criticism, the revised manuscript includes more mechanistic insights into the origin of nuclear envelope ruptures and more data from primary hippocampal neurons. In particular, we have addressed the reviewer's question whether PolyQ74 acts directly on the nuclear membrane or indirectly through increased apoptosis or mitotic defects, as outlined below.

Major issues

1. *The methods indicate that the major experiments in figure 1, quantifying nuclear rupture frequency in U2OS cells, and figures 2 and 3, examining lamin organization, were performed during S-phase arrest. This increases both nuclear lamina disruptions and nuclear rupture frequency, potentially masking or enabling the phenotypes associated with polyQ74-NLS expression. In addition, cancer cells have known defects in nuclear envelope structure that make them a poor model for processes occurring in normal cells not under extensive mechanical force (e.g. squeezing through small pores). To understand the effect of polyQ-NLS on nuclear rupture and nuclear lamina organization in neurons the ideal platform would be a non-transformed cell line with minimal disruptions of the nuclear envelope.*

- We only used S-phase arrest for the quantification of rupture frequency in Figure 1, in line with earlier studies that used the same cell line for quantifying NE ruptures (Hatch et al., 2013; Hatch & Hetzer, 2016; Vargas et al., 2012). In these experiments, the rupture frequency of cells expressing polyQ74-NLS aggregates was compared with the rupture frequency of cells expressing polyQ23-NLS and polyQ74. We found that expression of polyQ74-NLS aggregates strongly increased the rupture frequency ($32.9 \pm 8.6\%$, versus $7.2 \pm 1.9\%$ and $7.7 \pm 8.3\%$ in cells expressing polyQ23-NLS or polyQ74, respectively). These results were then confirmed in primary (post-mitotic) hippocampal neurons, directly demonstrating that nuclear polyglutamine aggregates rupture the nuclear envelope. We apologize that our methods section was unclear and raised the impression that S-phase arrest was also used for Figures 2 and 3, which is not the case. For determining lamina defects in Figure 2 and 3, cells that appeared to divide (e.g. due to condensed chromatin) were excluded from the analysis.

Because transient expression of NLS-RFP results in variable expression and increased nuclear ruptures, we used U2OS cells with stable expression of NLS-RFP to carefully quantify the rupture frequencies and repair probabilities in different conditions. To address the reviewer's concern related to this cancer cell line, our revised manuscript includes more data from primary, non-transformed and post-mitotic neurons, demonstrating impaired repair after rupture in neurons with

nuclear polyQ aggregates. Furthermore, Figure S2D demonstrate that aggregate-induced ruptures also occur in non-transformed RPE cells.

2. *A direct function of polyQ74-NLS in rupturing the membrane through induced curvature is not completely clear from the presented data. Although the images of polyQ74-NLS induced membrane curvature in Figure 3 are fantastic, the data in figure 1F and the figure 1 videos suggest that most of the ruptures in these cells occur in membrane blebs from which aggregates are excluded. In addition, it is likely that polyQ74 aggregates can only affect membrane curvature at lamina gaps. Thus the biological significance of the ability of polyQ74 aggregates to induce membrane curvature is unclear.*
 - We now realize that our discussion on the rupture mechanism was somewhat ambiguous and we have addressed this in the revised version. The reviewer correctly notes that ruptures are often preceded by blebs. These blebs always emerge in close proximity of polyQ74 aggregates and we believe that they are due to the aggregate-induced destabilization of the interaction between chromatin, the nuclear lamina and the nuclear envelope. We have now quantified these observations and found that 42% of rupture events were preceded by blebbing (Fig. 1, G), while other ruptures occur without visible preceding bleb formation. In combination with our result on aggregate-induced lamin destabilization, we now propose that blebbing and ruptures emerge as a result of local disruption of the chromatin-lamin-NE interaction in combination with compressive forces on the nucleus. We therefore tested the effect of blebbistatin-induced actomyosin inhibition and found that this indeed decreases the number of ruptures. We have included these results in the revised manuscript (Fig. 3), together with data that provide additional evidence that blebs are the sites of ruptures (i.e. combined RFP-NLS and cGAS imaging, Fig. 2, D and E).
3. *Expression of polyQ74-NLS is correlated with increased nuclear lamina disruption, but it is unclear how direct this is. One likely explanation is that the protein interferes with nuclear envelope assembly during mitosis. This could be determined by live-cell imaging or more clearly by inducing expression in interphase only. If this is mainly a mitotic phenotype in U2OS cells, it is unclear how the data in this model relates to the situation in neurons. From the videos the nuclei also appear very stiff, as if lamin A levels or chromatin organization were severely altered, which could also drive rupture indirectly.*
 - To address this point, we have added more data from primary, non-transformed and post-mitotic neurons, demonstrating impaired repair after rupture in neurons with nuclear polyQ aggregates (Fig. 5, C-G). In addition, it is important to point out that cells that transiently express polyQ74 rarely display cell divisions, which makes it unlikely that our results are mostly due to impaired nuclear envelope assembly. Interesting, we now also created a stable line with inducible expression of polyQ74. These cells show more cell divisions in the presence of polyQ aggregates, but also revealed that aggregates are kept out of the nucleus during nuclear envelope reformation. Thus, cells that exclusively show nuclear aggregates most likely did not have mitosis in the presence of aggregates.
4. *A direct function of polyQ74-NLS in preventing membrane repair is not clear from the presented data. This argument rests on the observation that a proportion of ruptured nuclei with polyQ74-NLS do not repair and that polyQ-aggregates are present at rupture sites and do not co-localize with lamin A. An alternative hypothesis is that nuclei with polyQ74-NLS aggregates are more likely to undergo apoptosis, which causes persistent nuclear rupture prior to cell death. Analysis of the timing between rupture and apoptosis in polyQ74 cells could be informative, as cells can persist with ruptured nuclei (caused by BAF depletion) for many hours prior to death, as well as IF analysis of apoptotic marks, such as lamin phosphorylation in cells with ruptured nuclei.*
 - In our original manuscript, we showed that in control cells or cells expressing polyQ74 or polyQ23-NLS ruptures are rapidly repaired, while in cells expressing polyQ74-NLS about half of the ruptures are not healed. This demonstrates that the presence of polyQ74 aggregates in the nucleus negatively

affects membrane repair. We furthermore present example images in which the fibrils formed by nuclear polyQ74 are pointing from the nucleus into the cytosol, which provides a straightforward mechanistic explanation for why the ESCRT complex will be unable to seal the membrane at these positions.

Over the past months, we have now also directly studied ESCRT-III and other components involved in nuclear envelope repair (BAF, emerin, LAP2B). This revealed that these factors accumulate near nuclear aggregates close to the nuclear envelope (Fig. 4, H-K), consistent with our model that fibrils impair repair because the ESCRT machinery stalls and cannot seal the membrane.

To address the reviewer's alternative suggestion that induction of apoptosis might underly unhealed ruptures in polyQ74-NLS expressing cells, we have now used a live-cell marker for mitochondrial membrane potential to analyse whether ruptures occur independent from apoptosis induction or not. This revealed that in most cells with unhealed NE ruptures, mitochondrial membrane potential is maintained. We have included these data into the revised manuscript (fig. S1, K-M).

5. *With regards to the potential mechanism, their evidence that polyQ74-NLS inhibits membrane recruitment is unclear. Both lamin A and cGAS bind chromatin (indirectly, in lamin A's case), not polyQ, and therefore their exclusion from aggregates is not unexpected. However, the images in Figure 2A, D, and E don't show the depletion depicted in the line graphs, making the biological significance of this observation unclear. In addition, their data in Figure 3 show a strong accumulation of membrane at a rupture site, suggesting that membrane can be recruited. To determine whether resealing is prevented additional analysis of rupture sites by correlative EM would be required and analysis of ESCRT-III protein localization by IF.*

Indeed, Figure 3 (now Fig. 4, F) shows a strong accumulation of membrane at a rupture site, which we interpret as evidence that ESCRT dependent repair is initiated but cannot be successfully completed (i.e. membrane can be added but sealing is obstructed by the fibrils. As mentioned above, we have now performed various experiments to study ESCRT-III and other components involved in NE repair at rupture sites that have aggregates present. We found that ESCRT-III component CHMP4B accumulates at nuclear aggregates close to the NE and we interpret this as evidence for stalled repair (Fig. 4, I).

Minor issues

1. *It is not clear initially from the text what the disease-relevance of the polyQ74-NLS construct is.*
 - Analysis of tissue from HD patients and mouse models has revealed that polyglutamine aggregates are mostly present in the nucleus of neurons. As such, polyQ74-NLS provides a more relevant context for HD than cytosolic polyQ. In post-mitotic neurons, polyQ74 without NLS also results in nuclear aggregates and the revised manuscript will include more data from this model system.
2. *Images of the nuclear lamins in Figure 2 are compromised by the use of Prolong Diamond mounting medium. This squashes the nucleus and gives rise to an aberrant intranuclear signal. In general, immunofluorescence analysis of nuclear envelope proteins requires mounting in a glycerol based medium. This phenomenon likely doesn't affect the authors' quantification of deformed nuclei, but it can obscure nuclear lamina disruptions by compressing the signal and makes interpretation of the images challenging.*
 - We thank the reviewer for pointing out the potential differences that could be found when using different mounting media. We have now compared the two mounting methods and found that, despite small differences in the resulting images, the phenotypes assessed in our quantification were indeed unaffected by the use of Prolong Diamond (see Supporting Figure 1). For consistency with our earlier data sets, we decided to continue the use of Prolong Diamond for this study.

Supporting Figure 1. Assessment of aggregate induced laminB1 phenotypes using a glycerol-based mounting medium. Representative images of U2OS WT cells expressing polyQ74-NLS or polyQ74 aggregates stained with LaminB1 and mounted using vectashield antifade mounting medium (Vector Laboratories; ref: H-1000-10). Orange arrowhead indicates major lamina disruptions in polyQ74-NLS expressing cells similar to those found in Fig. 2 (Fig. 2, H and J). Orange arrows indicate continuous lamina that is locally deformed close to a cytosolic aggregate (Fig. 2, H and I). Scale bar indicates 10 μ m.

3. Figure II: an additional graph with the half-times for the individual recovery curves would be beneficial to determine whether there are multiple repair types present in the different groups.

➤ We have included a graph with the time to half recovery in Figure 1 (panel G).

Reviewer #2

This study by Korsten et al reveals new cellular impacts of Huntington's disease-associated polyglutamine inclusions. These aggregates are shown to cause nuclear rupture, hinder repair mechanisms and lead to dysregulation of the nuclear lamina. Through live-cell and TREx imaging, the authors track nuclear rupturing by an NLS-directed polyQ74 construct, which results in nuclear cargo escape, cGAS response, and lamin deformation. This study builds upon previous results from Baurlein et al (11) demonstrating that polyQ aggregates form fibrils that deform membranes, as well as Gasset-Rosa et al (7) and Grima et al (8) highlighting how polyQ aggregates compromise NPC function and nucleocytoplasmic transport. Significantly, the present study reports that polyQ aggregates directly initiate membrane rupture and impede repair by extension of fibrils through the rupture site. The authors also show disruptions to the nuclear lamina that are reminiscent of findings by Denais et al (19) describing membrane rupture and repair during cell migration. Overall, we find the experimental approaches and interpretations of results sound, making this manuscript a valuable contribution to further understanding the pathology of Huntington's disease. We have a few concerns that should be addressed experimentally, with the main concern centering around the artificially created polyQ74-NLS construct and whether it recapitulates the effects of disease-associated variations.

- We are grateful to the reviewer for their positive feedback and constructive comments, which we have addressed in the revised version.

Major concerns

1. *Can the authors rule out that appending the highly charged NLS changes the properties of polyQ74 that induce nuclear rupture? This construct is observed to localize within the nucleus leading to inclusions with more fibrillar morphology compared to non-NLS bearing variants. Is there any evidence (for non-NLS bearing variants) that fibrillar structures analogous to polyQ74-NLS reach/breach the nuclear envelope? Do any of the cytosolic, perinuclear inclusions similarly breach the NE from the outside? Perhaps the authors could consider using a system that involves cleaving the NLS after nuclear entry via nuclear TEV, previously used by Hartl and others. Lastly, it would be informative if the authors could test an alternative, longer (and more pathogenic) variant of huntingtin exon1.*
 - Importantly, neurons expressing polyQ74 without NLS also display nuclear aggregates, as well as numerous ruptures near such aggregates, arguing against a key role for the NLS in our non-neuronal experiments. In addition, high-resolution microscopy of cytosolic aggregates revealed that cytosolic aggregates are also highly fibrillar (Fig. S3, A and B). However, although these aggregates could contact the nuclear envelope, they did not breach it or triggered ruptures of the nuclear envelope. This supports our model in which ruptures are not directly triggered by fibrils that poke through the membrane, but instead by the aggregated-induced destabilization of the nuclear lamina. Upon rupture, fibrils that poke into the cytosol will hinder repair and result in long-term leakage and accumulation of repair factors. Given our observations in neurons with nuclear aggregates without NLS, in combination with time constraints, we decided to not pursue the elegant TEV-based approach suggested by the reviewer. Cloning and expression challenges that we encountered with longer polyQ variants precluded us from properly testing these variant at this point.
2. *How is the repair process initiated after a rupture occurs? Have the authors considered the potential involvement of the ESCRT pathway and, if so, whether disruption to this pathway (either by knockdown or expression of dominant negative factors) would hinder repair of rupture sites? Even a minimalistic subset of experiments within the chosen (short) publication format would elevate the manuscript mechanistically.*
 - Following the suggestion of the reviewer, we have studied ESCRT-III and other components involved in nuclear envelope repair (BAF, emerin, LAP2B). Live-cell imaging proved difficult due

to numerous overexpression artefacts associated with these proteins, but immunocytochemistry revealed that these factors accumulate near nuclear aggregates close to the nuclear envelope, consistent with our model that fibrils impair repair because the ESCRT machinery stalls and cannot seal the membrane (Fig. 4, G-K).

3. *Can the authors rationalize/discuss why polyQ74 is nuclear excluded in U2OS cells but not in the neuronal cells?*
 - We believe that the ability to undergo mitosis (up to a certain expression level) could be one explanation why polyQ74 does not readily accumulate in the nucleus of cycling cells. While cells that transiently overexpress polyQ74-NLS do not undergo mitosis during the time window used for live-cell imaging, we have also created a stable cell line that expresses this construct and continues to divide. Interestingly, aggregates present in the nucleus before mitosis become cytosolic upon nuclear envelope breakdown and remain so when the new nuclei are formed. Thus, mitosis is a good way to get rid of nuclear aggregates. This could explain why potential aggregates that form in the nucleus of polyQ74 expressing U2OS cells will not be sustained there, but instead become cytosolic after division. This mechanism does not work in post-mitotic neurons and could explain why these display nuclear aggregates. In addition, functional differences in the nuclear protein quality control system likely exist between U2OS cells and neurons, and these could potentially result in the inability to prevent nuclear aggregate formation. Finally, aggregation might also be affected by variations in endogenous proteins that could be prone to co-aggregate.
4. *For the experiments with neurons, it would be helpful if the authors could perform a Z-scan to better differentiate between nuclear and cytosolic signal. Moreover, in Fig.4B, A notable buildup of cGAS is observed in cases of P74Q overexpression. The aggregates formed under these conditions show a more compact morphology, unlike the fibril structure seen in P74Q-NLS. This leads to the question of how the compact P74Q aggregates (vs. fibrils observed in U2OS with NLS-PolyQ) contribute to nuclear envelope (NE) rupture.*
 - It is important to point out that we do not think that the fibrils directly poke holes in the NE. Instead, we propose that nuclear aggregates destabilize the interaction between chromatin, lamina, and NE, which is something that more compact aggregates could also do. We furthermore propose that polyQ fibrils often prevent proper resealing of the nuclear envelope. The revised manuscript now also included evidence for sustained loss of nuclear envelope integrity in neurons, both with polyQ74-NLS and polyQ74 (Fig. 5, C-G).

Minor concerns

1. *Line 23: "envelope" is misspelled as "envelop" in the one-sentence summary.*
 - We have corrected this.
2. *Is there a directionality to bleb formation comparing NLS/non-NLS variants? Comparing Fig. 1C with Fig. 4J, the nuclear bleb points outward and inward, respectively.*
 - We apologize for not better explaining our interpretation of this image. We do not think this is a bonafide inward bleb, but rather an accumulation of membrane resulting from inefficient NE repair that was obstructed by a polyQ fibril. The presence of cGAS indicates that a rupture has taken place at this location. We have clarified the representations with labels to aid interpretation of the data (Fig. 4, F).
3. *In Fig. 4J, why is cGAS present - is this immediately preceding rupture?*
 - See also point 2 above. The presence of cGAS indicates that a rupture has taken place at this location. To further validate cGAS as a readout for rupture, we have included data from experiments where cGAS and RFP-NLS are imaged together. This reveals that cGAS enters the nucleus directly after

RFP-NLS leaks out of the nucleus and often appears at the site where blebbing preceded the rupture (Fig. 2, D and E).

4. *Based on approximately 20% of cells in Fig. 4B, is it technically feasible to identify polyQ74 fibrils in neurons? This would nicely support the data in U2OS cells.*
 - As anticipated by the reviewer, the limited number of cells made it challenging to perform Expansion Microscopy, which has a very low throughput and moderate success rate – especially for achieving proper labeling densities for aggregates. Nonetheless, we have now obtained results that reveal very clear polyQ74 fibrils in the cytosol of neurons (fig. S3, K). These fibrillar aggregates show a striking resemblance to (nuclear) polyQ aggregates in U2OS cells (fig. S3, A-B)
5. *Did the authors consider using DNA leakage as a readout? In Fig. 2A or other rupture events, the authors can attempt to stain DNA to determine whether cytoplasmic exposure of DNA occurs.*
 - The rapid and massive accumulation of cGAS following a rupture is a clear indication that the chromatin gets exposed to the cytosol.
6. *Did the authors normalize protein expression levels across constructs (e.g. via immunoblot).*
 - No, we did not do this because the expression is very variable per cells.
7. *Can LaminA accumulation be used as a score of rupture severity (as in citation 19) and related to polyQ74 aggregate size and/or frequency?*
 - We performed the suggested analysis and found no clear correlation between rupture severity and Lamin scar intensity. While this could reflect reduced scar formation due to the presence of aggregates, this analysis has some limitations. For example, in Denais *et al.*, scar formation takes place at the leading edge of the migrating nucleus and can therefore be robustly imaged. In our cells, it is often unclear around which aggregate a rupture will take place and lamin scars can emerge either in or (partially) out of the focal plane. We therefore did not further pursue this.
8. *polyQ74 protein is mostly referred to as polyQ74 and on three occasions as polyQ74cyto. Either name is fine as long as defined and used consistently.*
 - We thank for the careful reading and have made sure to consistently name the protein as “polyQ74”. We have replaced all instances where we previously used “polyQ74cyto”.
9. *Fig. 3J shows a magnified subfigure of polyQ74-NLS and mCLING staining that lacks cGAS staining, despite a box indicating magnification placed around the unzoomed image of cGAS and mCLING. This magnified image should include the cGAS staining to better resolve its localization around fibrils and the nuclear deformation site.*
 - In the revised version we have modified the display of the data (now in Fig. 4, F) with an additional zoomed in merge of mCLING and cGAS to aid visibility.
10. *Are there any instances of TREx imaging showing cGAS accumulation and/or mCLING absence at rupture sites? Fig 3J shows cGAS present as a marker for nuclear rupture, yet the image lacks other signs of rupture, such as a breach in mCLING staining and/or the protrusion of fibrils through the membrane. Because mCLING does not differentiate among membranes, the authors should clarify the nature of this inward blebbing. Showing rupture-localized cGAS under conditions such as those shown in Fig. 3F would help validate cGAS as a readout for rupture.*
 - See also point 2 and 3 above. We apologize for not better explaining our interpretation of this image. We do not think this is a bona fide inward bleb, but rather an accumulation of membrane resulting from inefficient NE repair that was obstructed by a polyQ fibril. To further validate cGAS as a readout for rupture, we will include data from experiments where cGAS and RFP-NLS are imaged together. This reveals that cGAS enters the nucleus directly after RFP-NLS leaks out of the nucleus and often appears at the site where blebbing preceded the rupture.

June 22, 2024

RE: JCB Manuscript #202307142R-A

Prof. Lukas Kapitein
Utrecht University
Padualaan 8
Utrecht 3533 CH
Netherlands

Dear Prof. Kapitein,

Thank you for submitting your revised manuscript entitled "Nuclear poly-glutamine aggregates rupture the nuclear envelope and hinder its repair." We would be happy to publish your paper in JCB pending final revisions necessary to address the remaining reviewer comments and to meet our formatting guidelines (see details below).

You will see that the reviewers ask for text changes to expand discussion of caveats and alternative models. If you are able to address any of the comments with new data that would be welcome as this will enhance the paper, but that is not required.

A. MANUSCRIPT ORGANIZATION AND FORMATTING:

1) Text limits: Character count for Reports is < 20,000, not including spaces. Count includes title page, abstract, introduction, results, discussion, and acknowledgments. Count does not include materials and methods, figure legends, references, tables, or supplemental legends.

2) Figure formatting: Reports may have up to 5 main text figures. Scale bars must be present on all microscopy images, including inset magnifications. Please add scale bars to Figure S1K as well as to magnification images in Figures 2C, 4F, S1B, S2A/E/G, & S3J.

Also, please avoid pairing red and green for images and graphs to ensure legibility for color-blind readers. If red and green are paired for images, please ensure that the particular red and green hues used in micrographs are distinctive with any of the colorblind types. If not, please modify colors accordingly or provide separate images of the individual channels.

3) Statistical analysis: Error bars on graphic representations of numerical data must be clearly described in the figure legend. The number of independent data points (n) represented in a graph must be indicated in the legend. Please, indicate whether 'n' refers to technical or biological replicates (i.e. number of analyzed cells, samples or animals, number of independent experiments). If independent experiments with multiple biological replicates have been performed, we recommend using distribution-reproducibility SuperPlots (please see Lord et al., JCB 2020) to better display the distribution of the entire dataset, and report statistics (such as means, error bars, and P values) that address the reproducibility of the findings.

Statistical methods should be explained in full in the materials and methods. For figures presenting pooled data the statistical measure should be defined in the figure legends. Please also be sure to indicate the statistical tests used in each of your experiments (both in the figure legend itself and in a separate methods section) as well as the parameters of the test (for example, if you ran a t-test, please indicate if it was one- or two-sided, etc.). Also, if you used parametric tests, please indicate if the data distribution was tested for normality (and if so, how). If not, you must state something to the effect that "Data distribution was assumed to be normal but this was not formally tested."

4) Materials and methods: Should be comprehensive and not simply reference a previous publication for details on how an experiment was performed. Please provide full descriptions (at least in brief) in the text for readers who may not have access to referenced manuscripts. The text should not refer to methods "...as previously described."

5) For all cell lines, vectors, constructs/cDNAs, etc. - all genetic material: please include database / vendor ID (e.g., Addgene, ATCC, etc.) or if unavailable, please briefly describe their basic genetic features, even if described in other published work or gifted to you by other investigators (and provide references where appropriate). Please be sure to provide the sequences for all of your oligos: primers, si/shRNA, RNAi, gRNAs, etc. in the materials and methods. You must also indicate in the methods the source, species, and catalog numbers/vendor identifiers (where appropriate) for all of your antibodies, including secondary. If

antibodies are not commercial, please add a reference citation if possible.

6) Microscope image acquisition: The following information must be provided about the acquisition and processing of images:

- a. Make and model of microscope
- b. Type, magnification, and numerical aperture of the objective lenses
- c. Temperature
- d. Imaging medium
- e. Fluorochromes
- f. Camera make and model
- g. Acquisition software
- h. Any software used for image processing subsequent to data acquisition. Please include details and types of operations involved (e.g., type of deconvolution, 3D reconstitutions, surface or volume rendering, gamma adjustments, etc.).

7) References: There is no limit to the number of references cited in a manuscript. References should be cited parenthetically in the text by author and year of publication. Abbreviate the names of journals according to PubMed.

8) Supplemental materials: Reports may have up to 3 supplemental figures and 10 videos. Please also note that tables, like figures, should be provided as individual, editable files. A summary of all supplemental material should appear at the end of the Materials and methods section. Please include one brief sentence per item.

9) Video legends: Should describe what is being shown, the cell type or tissue being viewed (including relevant cell treatments, concentration and duration, or transfection), the imaging method (e.g., time-lapse epifluorescence microscopy), what each color represents, how often frames were collected, the frames/second display rate, and the number of any figure that has related video stills or images.

10) eTOC summary: A ~40-50 word summary that describes the context and significance of the findings for a general readership should be included on the title page. The statement should be written in the present tense and refer to the work in the third person. It should begin with "First author name(s) et al..." to match our preferred style.

11) Conflict of interest statement: JCB requires inclusion of a statement in the acknowledgements regarding competing financial interests. If no competing financial interests exist, please include the following statement: "The authors declare no competing financial interests." If competing interests are declared, please follow your statement of these competing interests with the following statement: "The authors declare no further competing financial interests."

12) A separate author contribution section is required following the Acknowledgments in all research manuscripts. All authors should be mentioned and designated by their first and middle initials and full surnames. We encourage use of the CRediT nomenclature (<https://casrai.org/credit/>).

13) ORCID IDs: ORCID IDs are unique identifiers allowing researchers to create a record of their various scholarly contributions in a single place. Please note that ORCID IDs are required for all authors. At resubmission of your final files, please be sure to provide your ORCID ID and those of all co-authors.

14) Journal of Cell Biology now requires a data availability statement for all research article submissions. These statements will be published in the article directly above the Acknowledgments. The statement should address all data underlying the research presented in the manuscript. Please visit the JCB instructions for authors for guidelines and examples of statements at (<https://rupress.org/jcb/pages/editorial-policies#data-availability-statement>).

B. FINAL FILES:

****The license to publish form must be signed before your manuscript can be sent to production. A link to the electronic license to publish form will be sent to the corresponding author only. Please take a moment to check your funder requirements before choosing the appropriate license.****

Thank you for your attention to these final processing requirements. Please revise and format the manuscript and upload materials within 14 days. If you need an extension for whatever reason, please let us know and we can work with you to determine a suitable revision period.

Thank you for this interesting contribution, we look forward to publishing your paper in Journal of Cell Biology.

Sincerely,

Martin Hetzer, PhD
Monitoring Editor
Journal of Cell Biology

Dan Simon, PhD
Scientific Editor
Journal of Cell Biology

Reviewer #1 (Comments to the Authors (Required)):

The current version solidifies the data that polyQ74 can induce NE ruptures in a disease relevant context, but the mechanism of the rupture and repair defects are still not defined. It is clear that polyQ74-NLS can induce nuclear lamina defects in U2OS cells, unclear in neurons, but evidence for direct disruption of the nuclear lamina through interactions with the aggregate is slim. Similarly, evidence that repair protein recruitment is disrupted in the presence of aggregates is missing and the membrane phenotype is not characterized well enough to say whether it is unusual.

Overall, I would characterize this paper as observational with an interesting, but untested, model for how aggregates could induce NE rupture and delay repair.

1. Page 6, line 125: "Nuclear aggregates locally disrupt the nuclear lamina".

The major support for this spatiotemporal interpretation is a single observation of a growing aggregate coinciding with a decrease in lamin A fluorescence (Fig. S2E-F). Without quantifying how frequently lamina disruption coincides with formation or association of an aggregate at the NE, it is equally likely that the aggregates induce nuclear lamina disruption by globally disrupting chromatin structure, nuclear mechanics, or nuclear trafficking in addition to or instead of a local action on the NE. The section should therefore include a discussion of this alternative model.

The relationship to the membrane blebs is also still unclear since none of the examples of aggregates at rupture sites appear to have membrane blebs (by RFP-NLS). Could there be two different mechanisms of rupture in these cells?

2. Page 9, line 200: "Notably, these results not only suggest higher rupture frequencies induced by nuclear aggregates, consistent with our live-cell observations of ruptures in U2OS-RFP-NLS cells, but might also reflect ESCRT-III stalling at rupture sites."

Page 10, line 213: "and the accumulation of repair factors at rupture sites indicates that nuclear polyQ aggregates indeed interfere with proper membrane resealing after NE rupture."

It is clear that there are nuclear membrane repair defects in polyQ74-NLS U2OS cells since many of them eventually fail to repair at all. However, the data presented suggest that repair proteins are recruited normally to rupture sites. As they mention, their data are consistent with a higher rupture frequency in polyQ74-NLS cells. No data is presented that indicates that repair

protein recruitment, including Chmp4B, is altered compared to ruptures of a similar size in cells without aggregates. The observation of membrane folding (mCLING) at polyQ74-NLS rupture sites is interesting, but it is not clear how frequently they occur in these cells versus controls and how frequently they co-localize with aggregates. Therefore, these sentences should be reworked to clarify what is an interpretation of the data and what are potential hypotheses for future investigation.

3. Page 11, line 250: "Together these findings demonstrate that nuclear aggregate-induced NE ruptures and long-term loss of NE integrity also occur in the cell type primarily affected in HD."

Since the authors cannot look at rupture duration or frequency directly and these nuclei are known to have nuclear transport defects, they cannot exclude that persistent cytoplasmic RFP-NLS is the result of a single rupture plus impaired reimport, similar to what is observed with nuclear import inhibitor drugs. The section should therefore include a discussion of this alternative model.

Reviewer #2 (Comments to the Authors (Required)):

Overall, the authors addressed our concerns in minimal-essential fashion. They provide new data addressing our concern with the NLS. While alternate polyQ variants were not tested, the issue of distinct polyQ deposition in mitotic vs. post-mitotic was addressed. We suggest to include the rationale provided in this context in the rebuttal (related to our prior major concern#3) also in the main text of the manuscript, as we feel this is an important point of discussion for the readership (we do not need to see these revisions again as they do not involve new experiments).

All in all, the manuscript reports on an interesting phenomenon and is in our opinion acceptable for a short format in JCB.

Utrecht, July 10, 2024

Re: *Nuclear poly-glutamine aggregates rupture the nuclear envelope and hinder its repair*

Dear Dan,

Thank you so much for provisionally accepting our manuscript. We are now resubmitting a final version following the editorial instructions. To address the remaining reviewer comments we have made several changes to the text. Specifically:

- We now state that lamina disruption could also be caused by global disruption of chromatin structure, nuclear mechanics, or nuclear trafficking (line 133, comment 1 of reviewer 1).
- We have slightly altered the wording to emphasize that we do not provide definitive proof for repair factor stalling (line 224, comment 2 of reviewer 1).
- We have added a sentence discussing that the lack of nuclear enrichment observed in neurons could also be due to impaired nucleocytoplasmic shuttling (line 265, comment 3 of reviewer 1).
- We have added a brief discussion on polyQ deposition in mitotic versus post-mitotic cells (line 241, comment 1 of reviewer 2).

Thank you very much for your excellent editorial guidance. We are excited about the prospect of publishing our work in Journal of Cell Biology and look forward to seeing our manuscript in print.

Sincerely yours,

Lukas Kapitein
Professor of Molecular and Cellular Biophysics
Utrecht University, The Netherlands
l.kapitein@uu.nl